# *C9ORF72*-derived poly-GA DPRs undergo endocytic uptake in iAstrocytes and spread to motor neurons

Paolo M Marchi[1,3], Lara Marrone[1,3], Laurent Brasseur[2], Audrey Coens[2], Christopher P Webster[1,3], Luc Bousset[2], Marco Destro[1,3], Emma F Smith[1,3,4], Christa G Walther[5], Victor Alfred[1,3], Raffaele Marroccella[1,3], Emily J Graves[1,3], Darren Robinson[5], Allan C Shaw[1,3], Lai Mei Wan[1,3], Andrew J Grierson[1,3], Stephen J Ebbens[6], Kurt J De Vos[1,3,4], Guillaume M Hautbergue[1,3], Laura Ferraiuolo[1,3], Ronald Melki[2], Mimoun Azzouz[1,3]

Dipeptide repeat (DPR) proteins are aggregation-prone poly-peptides encoded by the pathogenic GGGGCC repeat expansion in the *C9ORF72* gene, the most common genetic cause of amyotrophic lateral sclerosis and frontotemporal dementia. In this study, we focus on the role of poly-GA DPRs in disease spread. We demonstrate that recombinant poly-GA oligomers can directly convert into solid-like aggregates and form characteristic β-sheet fibrils in vitro. To dissect the process of cell-to-cell DPR transmission, we closely follow the fate of poly-GA DPRs in either their oligomeric or fibrillized form after administration in the cell culture medium. We observe that poly-GA DPRs are taken up via dynamin-dependent and -independent endocytosis, eventually converging at the lysosomal compartment and leading to axonal swellings in neurons. We then use a co-culture system to dem-onstrate astrocyte-to-motor neuron DPR propagation, showing that astrocytes may internalise and release aberrant peptides in disease pathogenesis. Overall, our results shed light on the mechanisms of poly-GA cellular uptake and propagation, sug-gesting lysosomal impairment as a possible feature underlying the cellular pathogenicity of these DPR species.

## Introduction

The polymorphic hexanucleotide repeat expansion (HRE) in the *C9ORF72* gene is the major genetic cause of amyotrophic lateral sclerosis/frontotemporal dementia (ALS/FTD) (DeJesus-Hernandez et al, 2011; Renton et al, 2011). The pathogenic HRE consists of hundreds to thousands of GGGGCC ($G_4C_2$) repeats, located in the first intron of the C9ORF72 gene (van Blitterswijk et al, 2013). A crucial driver of *C9ORF72*-mediated ALS/FTD pathology is the

unconventional repeat-associated non-AUG (RAN) translation of the HRE into five toxic dipeptide repeat (DPR) species: poly-PA, poly-GA, poly-PR, poly-GR, and poly-GP.

Among the five different DPRs generated by RAN translation, the most toxic species are considered to be arginine-containing ones, namely poly-GR and poly-PR. These DPRs have been shown to alter the formation of membrane-less organelles such as stress granules or nucleoli (Tao et al, 2015; Lee et al, 2016; Lin et al, 2016; Zhang et al, 2018), cause mitochondrial dysfunction and DNA damage (Lopez-Gonzalez et al, 2016; Choi et al, 2019), and their expression is toxic in mice and in iPSC-derived cortical and motor neurons (Lopez-Gonzalez et al, 2016; Cook et al, 2020). However, poly-GA DPRs appear to be the most abundantly detected DPR species (Zhang et al, 2014; Mackenzie et al, 2015), and their toxicity has been docu-mented both in cell culture and in vivo (Lee et al, 2017; Ohki et al, 2017; Nihei et al, 2020), correlating with motor deficits, cognitive defects and inflammatory response in mice (Zhang et al, 2016; Schludi et al, 2017; LaClair et al, 2020). *Post-mortem* tissue of ALS/FTD patients contains ubiquitin- and p62-positive DPR inclusions predominantly in the frontal cortex, hippocampus and cerebellum of neuronal cells (Ash et al, 2013; Mori et al, 2013; Schludi et al, 2015; Saberi et al, 2018), with rare occurrence in glia (Rostalski et al, 2019). Initial efforts to identify molecular mechanisms of poly-GA toxicity revealed that this DPR species interacts with components of the Ubiquitin-Proteasome System, such as p62, ubiquilin-1, ubiquilin-2, HR23 (May et al, 2014; Schludi et al, 2015; Zhang et al, 2016) and specifically leads the 26S proteasome to stalled degradation (Guo et al, 2018). Emerging evidence shows that poly-GAs can also rapidly spread throughout the Drosophila brain in a repeat length- and age-dependent manner (Morón-Oset et al, 2019), in agreement with the ability of poly-GAs to spread and drive cytoplasmic mislocalization and aggregation of TDP-43 in cell cultures (Chang et al, 2016; Westergard et al, 2016; Zhou et al, 2017; Khosravi et al, 2020).

[1]Sheffield Institute for Translational Neuroscience (SITraN), Department of Neuroscience, University of Sheffield, Sheffield, UK  [2]The French Alternative Energies and Atomic Energy Commission (CEA), Institut François Jacob (MIRcen) and The French National Centre for Scientific Research (CNRS), Laboratory of Neurodegenerative Diseases (UMR9199), Fontenay-aux-Roses, France  [3]Neuroscience Institute, University of Sheffield, Western Bank, Sheffield, UK  [4]Centre for Membrane Interactions and Dynamics, University of Sheffield, Western Bank, Sheffield, UK  [5]The Wolfson Light Microscopy Facility, University of Sheffield, Sheffield, UK  [6]Department of Chemical and Biological Engineering, University of Sheffield, Sheffield, UK

Correspondence: ronald.melki@cnrs.fr; m.azzouz@sheffield.ac.uk

In this work, we analyse poly-GA oligomers of 34 repeats. We first illustrate the process of poly-GA oligomer coalescence into solid-like species of amyloid nature with characteristic β-sheets in vitro. We next explore whether different poly-GA species (poly-GA oligomers vs poly-GA fibrils) may produce distinct features in terms of cellular uptake and cell-to-cell propagation. We show that poly-GA oligomers (and not fibrillary GA) enter cells despite dynamin inhibition, thus escaping lysosomal degradation.

Upon active uptake, both poly-GA oligomers and fibrils are transported to lysosomes, which become aberrantly enlarged and static, leading to axonal swellings in neurons. When astrocytes and motor neurons are co-cultured, DPR species are promptly internalised by astrocytes and spread to neuronal units. Our data highlight the steps of poly-GA DPR spread in cell culture, suggesting lysosomal impairment as a potential pathogenic mechanism underlying disease.

# Results

### Recombinant poly-GA aggregation into oligomeric and fibrillar assemblies

To study the potential role of poly-GA DPRs in ALS/FTD spread, we first set up an expression system in *Escherichia coli* for the production of recombinant poly-GA DPRs labelled with Atto-488, Atto-550, or Atto-647N dyes (see the Materials and Methods section; Fig S1). In addition, we included poly-PA DPRs for comparison. Immunoreactivity and fluorescent labelling of the generated GA/PA-repeat recombinant proteins were confirmed by dot-blotting or protein gel electrophoresis (Fig S2).

We first investigated the aggregation of poly-GA and -PA at different concentration in vitro (test-tube) using confocal microscopy. Both ATTO550-labelled DPRs coalesced into microscopic protein clusters under low-salt concentrations and without the addition of any molecular crowders (Fig 1A and B). Clusters formation was much faster for poly-PA than poly-GA, and exhibited increasingly larger size with increasing protein concentration (1, 10, and 20 μM) (Fig 1C) and incubation length (0, 2, 24 h) (Fig 1D) for both DPRs. The generated poly-GA and -PA clusters differed significantly in morphology and size, hence we used Z-stack confocal microscopy and CMLE deconvolution to visualize their 3D-volume and -surface rendering (Fig S3A and B). The 3D-reconstructed poly-GAs showed an irregular and compact solid-like structure (Video 1). In sharp contrast, the 3D-reconstructed poly-PAs were made up of very small spherical particles (circularity = 0.95; Fig S3C–E) resembling liquid droplets (Video 2). Using longer incubation times, we then aimed to investigate whether poly-GA oligomers could grow in vitro into β-sheet fibrils. As observed by transmission electron microscopy, we noticed that, whereas poly-PA formed non-fibrillary amorphous assemblies, poly-GA assembled into fibrillar structures within 15 d of incubation (Fig 1E and F). Notably, the poly-GA fibrillar DPRs used in this study were extensively characterized for their amyloid β-sheet content by Fourier-transform infrared spectroscopy (Brasseur et al, 2020). In summary, our poly-GA oligomers produced distinctive solid-like assemblies, nucleation

growth and 3D-architecture and uniquely assembled into characteristic β-sheet fibrils.

### Oligomeric and fibrillar poly-GA DPRs use distinct entry-routes in glia

Because poly-GA oligomers evolved into fibrillary assemblies in vitro, we aimed to compare the behaviour of these two species in cell culture by administering them directly into the culture medium. To evaluate DPR uptake, we exposed various cell types (HEK293T, HeLa, induced neural progenitor cell [iNPC]-derived human astrocytes, human fibroblasts) to 1 μM of our recombinant DPRs for 1, 2, and 4 h. Using high-throughput confocal microscopy, we then quantified the number of largely visible DPR aggregates in these cell types. Our results on ~8,000 cells/culture showed that poly-GA oligomers are taken up into large visible aggregates more readily than poly-GA fibrils and poly-PA oligomers in all the cell lines. In addition, for all DPR species and across all cell lines, the uptake increases with time (Fig 2A).

Because increasing evidence implicates astrocytes as significant non-cell autonomous contributors of *C9ORF72*-associated ALS/FTD pathogenesis (Varcianna et al, 2019), we aimed at better investigating poly-GA DPR uptake (at 1 μM) in healthy iNPC-derived human astrocytes (herein referred to as iAstrocytes). We first confirmed DPR internalisation in vimentin-stained iAstrocytes by using Z-stack AiryScan confocal microscopy, orthogonal views (Fig 2B) and 3D-volume rendering (Fig 2C and Videos 3 and 4). We further showed DPR uptake in glia by the use of 3D-STORM imaging (Fig 2D and E) and trypan blue quenching (Fig S4A), and DPR binding to cells by flow cytometry (Fig S4B) and anti-V5 dot-blot (Fig S4C and D). Uptake was also evidenced by live-cell imaging of iAstrocytes (Video 5) and subsequent quantification of DPR average velocity (*v*) with a mean square displacement (MSD) analysis of the *xy* trajectories of poly-GA aggregates (Fig 3A; see the Materials and Methods section). MSD analysis showed non-zero values for *v* (Fig 3B), which, according to our mathematical model, indicates that poly-GA motion is not solely due to random diffusion, but an active force is also contributing. Interestingly, our model suggests that this active force could be descriptive of microtubule-mediated transport, as reported in a previous study (van den Heuvel et al, 2007). To obtain the experimental confirmation, we exposed healthy iAstrocytes to 1 μM poly-GA (24 h) and subsequently subjected the cells to a 30 min-treatment with the microtubule de-polymerising agent nocodazole (30 μM). Cells were fixed and stained with anti-tubulin antibody to confirm microtubule de-polymerization. Our results show that the microtubule de-polymerising agent nocodazole induces cellular relocation of poly-GA DPRs (Fig 3C), hence suggesting that a certain fraction of poly-GAs undergoes microtubule-mediated transport after cell entry.

To test the potential involvement of endocytosis in DPR uptake, we performed a generalised block of all endocytic pathways by lowering the culture temperature to 4°C before fixation (Harding et al, 1983). To confirm the successful inhibition of endocytosis we visually monitored the uptake of transferrin, a well-established marker of the coated pit pathway (Ehrlich et al, 2004; Hanover et al, 1984). Interestingly, low-temperatures reduced the uptake of 1 μM poly-GA oligomers by 2.2-fold and of 1 μM poly-GA fibrils by ~8-fold

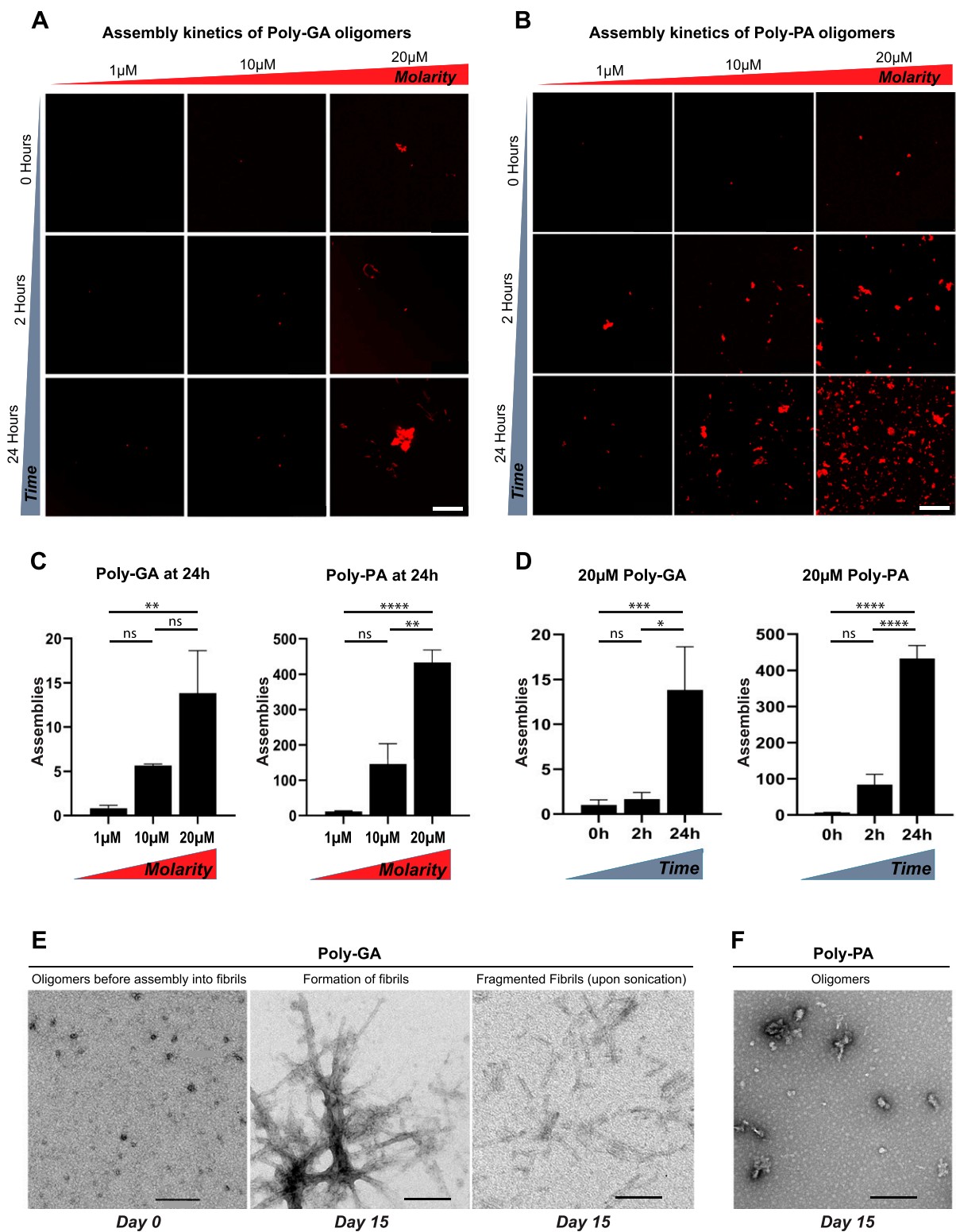

**Figure 1. Poly-GA oligomers form solid-like structures and assemble into characteristic β-sheet fibrils.**
**(A, B, C, D)** Aggregation of (A) poly-GA and (B) poly-PA oligomers in vitro, with relative quantification of the number of assemblies formed upon (C) increasing molarity or (D) time. Bar graphs of mean ± SEM. One-way ANOVA with Tukey's multiple-comparisons test. *$P \leq 0.05$, **$P \leq 0.01$, and ****$P \leq 0.0001$. The data were collected from two independent biological replicates. **(E, F)** Electron micrographs show that poly-GA oligomers form characteristic fibrils after 15 d in vitro (E), unlike poly-PA oligomers (F). **(A, B, E, F)** Poly-GA fibrils are shown before and after sonication, which drives the production of fragmented fibrils. Scale bar: 50 $\mu$m (A, B); 200 $\mu$m (E, F).

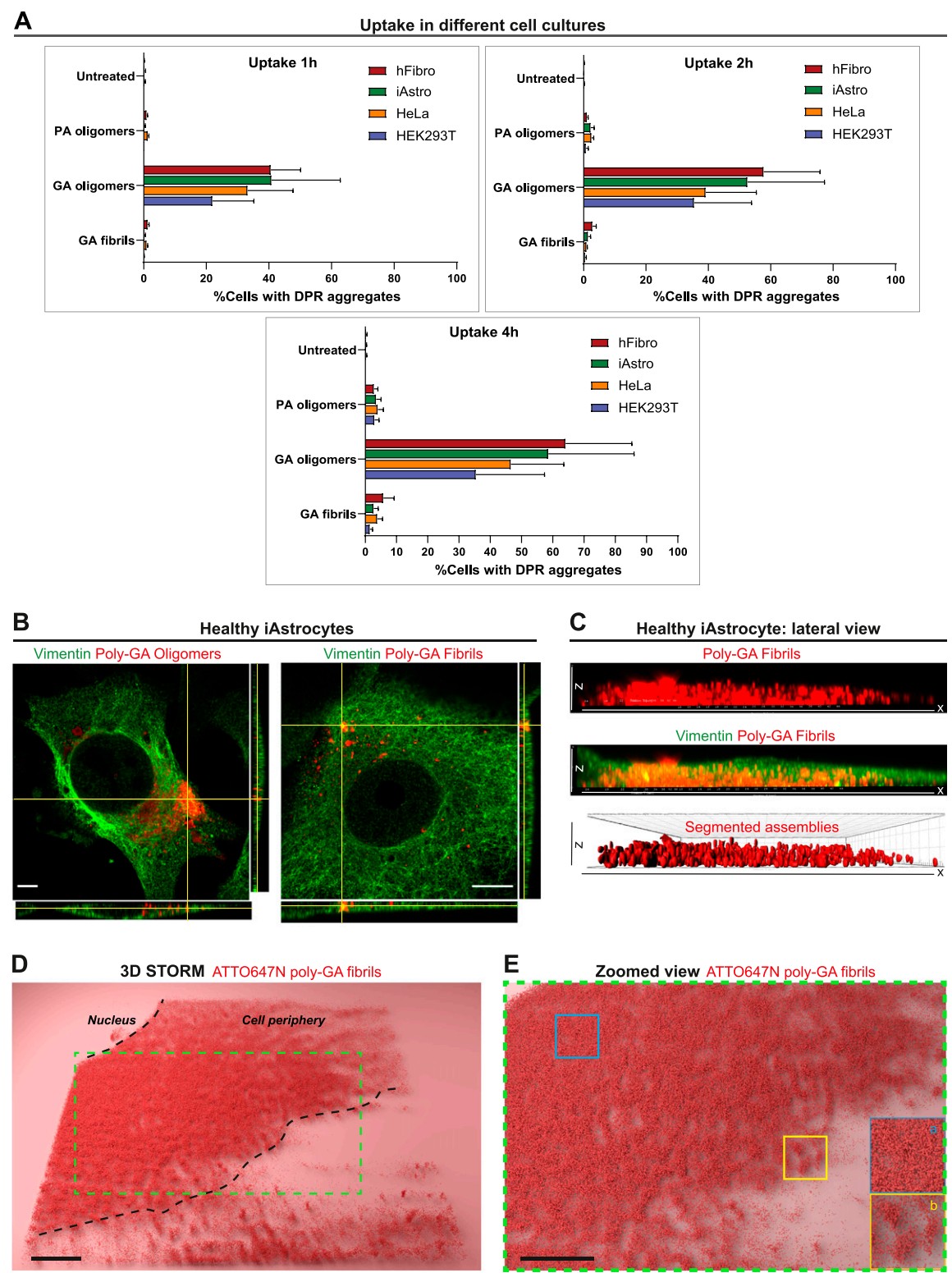

**Figure 2. Dipeptide repeat uptake in various cell lines and in healthy iAstrocytes.**
**(A)** Quantitative analysis of the %cells displaying large dipeptide repeat aggregates (1 μM) overtime (1 h, 2 h, 4 h). Different cell cultures were used in this experiment such as human fibroblasts (hFibro, red), iAstrocytes (iAstro, green), HeLa cells (orange), and HEK293T cells (blue). ~8,000 cells/culture were analysed. The data were collected from three independent biological replicates. Bar graphs of mean ± SEM. **(B)** Orthogonal views from AiryScan microscopy show the uptake of 1 μM poly-GA oligomers and 1 μM poly-GA fibrils in vimentin-stained healthy iAstrocytes after 24 h exposure. Scale bar = 10 μm. **(C)** 3D-rendered lateral view of a single vimentin-stained healthy iAstrocyte shows the uptake of 1 μM poly-GA fibrils through the xz dimension. Corresponding movies are shown in Videos 3 and 4. **(D)** 3D-STORM imaging

(****$P \leq 0.0001$) in iAstrocytes (Fig 4A). We next inhibited dynamin-dependent endocytosis by the use of dynasore (Macia et al, 2006; Kirchhausen et al, 2008), and this resulted in a 2.4-fold uptake reduction of poly-GA fibrils (****$P \leq 0.0001$) but no change in poly-GA oligomers uptake (Fig 4B). Upon comparison with other DPR oligomeric species (such as poly-PA), we observed that specifically poly-GA oligomers do not use dynamin-dependent endocytosis for cell entry. To better investigate the differences in uptake between oligomeric versus fibrillar poly-GA DPRs, we evaluated the accumulation of these proteins in endolysosomal organelles following 24 h from administration. We exposed healthy iAstrocytes to ATTO647N-labelled poly-GA fibrils or oligomers (0.5 μM for 24 h) and quantified the degree of colocalization with LAMP1-stained endolysosomes; with a lateral resolution of 50 nm for both channels (Fig S5A and B). The analysis revealed that whereas ~17% of the input poly-GA fibrils colocalized with endolysosomes, less than 5% poly-GA oligomers did (Fig 4C; 3.4-fold difference; ***$P \leq 0.001$). In addition, LAMP1 organelles showed higher enrichment for poly-GA fibrils than for poly-GA oligomers (1.6-fold difference; ***$P \leq 0.001$) (Fig 4C). Together, these findings suggest that DPR uptake is present in various cell culture systems and endocytosis plays a role in DPR uptake in iAstrocytes. However, there appears to be differences in DPR internalisation depending on the considered DPR species, with poly-GA oligomers being less reliant on dynamin-endocytosis than their fibril counterpart.

### Poly-GA DPRs accumulate into enlarged lysosomes and into axonal swellings in neurons

The finding that DPR colocalize with the endolysosomal compartment prompted us to assess whether they might be able to escape this compartment via lysosomal disruption, thus triggering cellular toxicity. This was performed using cortical neurons, which are known to undergo typical degeneration in ALS-FTD and display a large number of DPR inclusions in *post-mortem* tissue of ALS-FTD patients (Mackenzie et al, 2015).

After growing primary mouse cortical neurons into microfluidic culture chambers, we used live-imaging and detected poly-GA uptake in the soma (point of DPR exposure–at 1 μM) as well as in the proximal axon. The trajectories of poly-GA oligomers and poly-GA fibrils were then MSD analysed during live-cell imaging at the proximal axon compartment showing that poly-GA DPRs shuttle along axons by a mixture of diffusive, transported, and constrained motion (Fig S6A and B). Constrained motion was the predominant DPR motion throughout the axons; suggesting that these proteins are mostly anchored to static axonal structures. We indeed found that poly-GA DPRs accumulated in large axonal swellings, implying a failure of their axonal transport (Fig 5A). Interestingly, by zooming into few axonal swellings with higher resolution, we observed the presence of small poly-GA proteins erratically moving within each axonal swelling (Fig 5A-right and Video 6). We then used the lysosomal dye LysoTracker Green and observed that some of these

axonal swellings contain poly-GA colocalizing with lysosomal organelles (Fig S6C and Video 7). We next performed a thorough colocalization analysis between all the DPRs and the lysosomes contained in the proximal axons (Video 8). By using colour deconvolution algorithms, we could separately analyse two lysosomal populations: one which does not colocalize with poly-GA DPRs ("Non-Colocalized Lysosomes," NCLs) and one showing colocalization ("Colocalized Lysosomes," CLs). Interestingly, CLs displayed reduced displacement, reduced speed, and increased size compared with NCLs in proximal axonal regions (Fig 5B and C). Subsequently, we exposed primary cortical neurons to 1 μM poly-GA DPRs for 24 h and examined the transcriptional profile of various lysosomal metabolism genes. Whereas transcripts encoding various cathepsins (CTSL, CTSB, and CTSD), a lysosomal hydrolase (GBA), and a cation-permeable lysosomal channel (MCOLN1) did not show variation upon DPR exposure, the transcriptional levels of ATP6V0E1 (a component of the V-ATPase) were increased upon treatment with poly-GA fibrils (Fig S7A). The protein levels of cathepsins L, B, and D were then measured in neurons treated with 1 μM poly-GA DPRs for 24 h using immunofluorescence assays; each of these enzymes has previously been linked to the breakdown of misfolded protein aggregates (Cullen et al, 2009; McGlinchey & Lee, 2015; Wang et al, 2015). When compared with untreated neurons, the levels of cathepsins L, B, and D in neurons exposed to poly-GA showed no difference (Fig S7B). Using self-quenched enzymatic substrates (LysoSubstrate) that are targeted to lysosomes (Humphries & Payne, 2012), we then quantified in situ lysosomal enzyme activities. Neuronal lysosomes showed no difference in Lyso-Substrate levels after 24 h exposure to 1 μM poly-GA, indicating that poly-GA does not cause significant changes in lysosomal functionality (Fig S7C and D). We also used the galectin-3 immunofluorescence puncta assay (Aits et al, 2015) to look for lysosomal damage in DPR-exposed HeLa cells but found that galectin-3 levels in 1 μM DPR-exposed cells did not increase when compared with untreated cells (Fig S7E and G)—the lysosomotropic drug LLoMe was used as a positive control. Despite the lack of evident lysosomal malfunction in cells treated with poly-GA in these assays, we discovered that HeLa cells decreased their lysosomal pH after 24 h exposure to 1 μM poly-GA fibrils or oligomers, which corresponded to an increase in LysoSensor fluorescence (Fig S7F and H). This suggests that poly-GA may impair lysosomal acidification.

### Poly-GA DPRs undergo astrocyte-to-motor neuron spread

Despite neuron-to-astrocyte transmission has been documented (Westergard et al, 2016), the role of glial cells in DPR propagation remains mostly unexplored, and no study to our best knowledge has yet unveiled whether DPRs can propagate from astrocytes to neurons. To test the possibility of this directionality, we established a co-culture system between iAstrocytes and Hb9-GFP mouse motor neurons. Briefly, we treated healthy iAstrocytes with 1 μM DPRs for 24 h, performed a number of PBS washes to remove

---

was used to visualize 1 μM ATTO647N labelled poly-GA fibrils in healthy iAstrocytes after 24 h uptake. **(E)** The zoomed view shows how molecules of poly-GA fibrils can be densely clustered (inset a) but also compartmentalized in spherical structures (inset b).

## DPR uptake and trajectories in healthy iAstrocytes

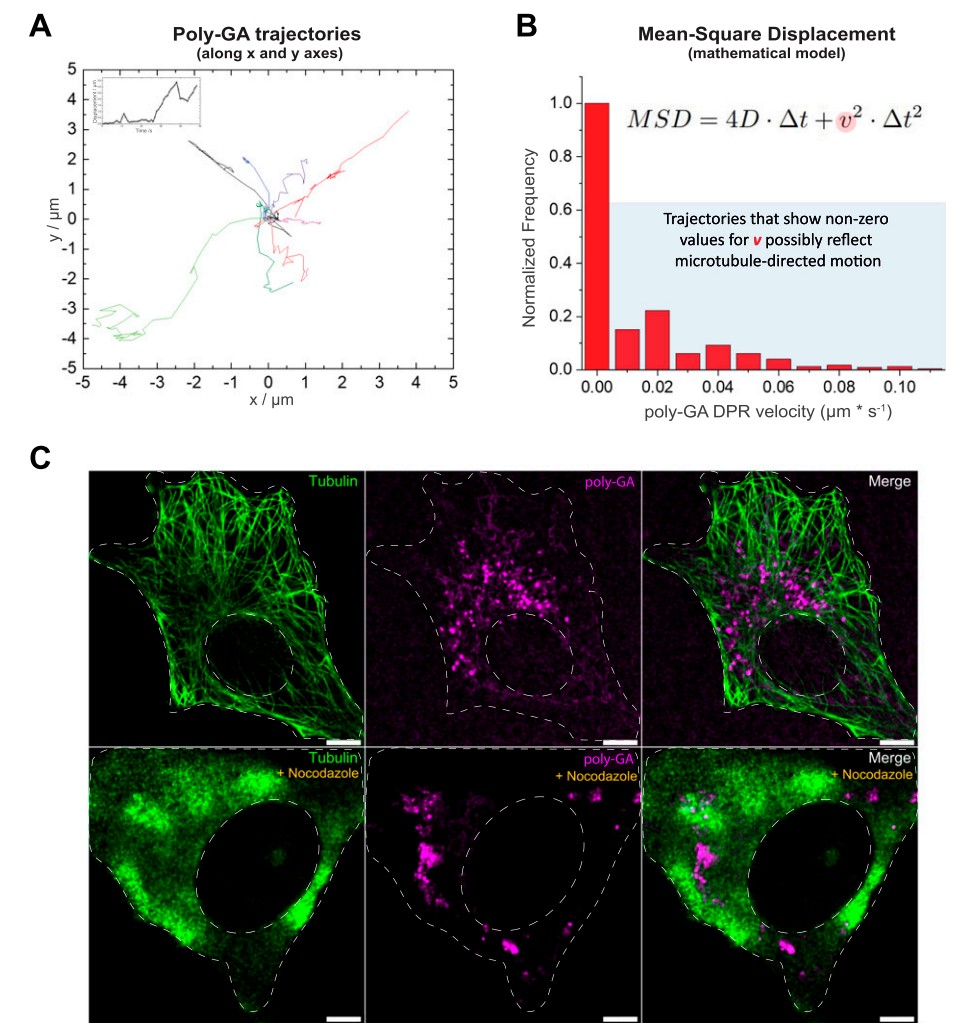

**A** Poly-GA trajectories
(along x and y axes)

**B** Mean-Square Displacement
(mathematical model)

$$MSD = 4D \cdot \Delta t + v^2 \cdot \Delta t^2$$

Trajectories that show non-zero values for **v** possibly reflect microtubule-directed motion

poly-GA DPR velocity (µm * s⁻¹)

**C**

Tubulin | poly-GA | Merge

Tubulin + Nocodazole | poly-GA + Nocodazole | Merge + Nocodazole

**Figure 3.   Dipeptide repeat (DPR) motion after uptake in healthy iAstrocytes.**
After 24 h exposure to 1 µM DPRs, healthy iAstrocytes were prepared for confocal live-imaging and ATTO550 DPR trajectories were analysed via mean Square Displacement. **(A)** Some tracks, analysed with TrackMate, are shown in the *xy* space. Corresponding movie shown in Video 5. **(B)** Quantification of DPR average velocity *v* is displayed in the corresponding graph, in which trajectories that show non-zero values for *v* are suggestive of microtubule-directed motion. **(C)** After 24 h exposure to 1 µM DPRs, healthy iAstrocytes were subjected to a 30-min pre-treatment with the microtubule de-polymerising agent nocodazole (30 µM) before fixation. Confocal images are shown, following anti-tubulin (green) immunostaining and ATTO550 fluorophore detection for DPRs (magenta). Nocodazole treatment induces cellular relocation of DPRs. Scale bar = 10 µm (upper panel), 5 µm (lower panel). The data were collected from two independent biological replicates.

remaining assemblies, and subsequently plated Hb9-GFP mouse motor neurons over the astrocyte layer. We kept this co-culture system for 48 h before fixation and confocal imaging (Fig 6A). We observed that, after astrocytic uptake, DPR assemblies underwent astrocyte-to-neuron propagation as confirmed by orthogonal views (Fig 6B). Different DPRs showed different percentages of propagation to motor neurons, with poly-GA fibrils being the most efficient at spreading (detected in 24% of motor neurons) when compared with oligomeric species (poly-GA: 4%; poly-PA: 7%) (Fig 6C). To confirm that iAstrocytes secrete DPRs, we performed an assay on single-cultured iAstrocytes in which we quantified DPR release into the conditioned medium (CM) by spectrophotometrically measuring ATTO550 fluorescence levels (Fig 6D). Eventually, we investigated whether the astrocyte-to-neuron

transmission of DPRs could contribute to motor neuron damage as a non-cell autonomous effect. However, no evident cytotoxicity was found in motor neurons upon co-cultures with iAstrocytes that contained and transmitted DPRs (Fig 6E). Furthermore, no increase in apolipoprotein J (APOJ), a lipoprotein secreted in lipoparticles that has been linked to astrocyte-induced toxicity (Guttenplan et al, 2021), was found in the CM of single-cultured iAstrocytes exposed to different DPRs at 1 µM. APOJ levels, as measured by ELISA, were actually reduced after poly-GP (and partially poly-GA) exposure, implying that DPRs may interfere with APOJ-mediated lipoparticle release (Fig S8A and B).

Taken together, our results show that iAstrocytes can efficiently release alanine-rich DPRs into the culture medium and, when

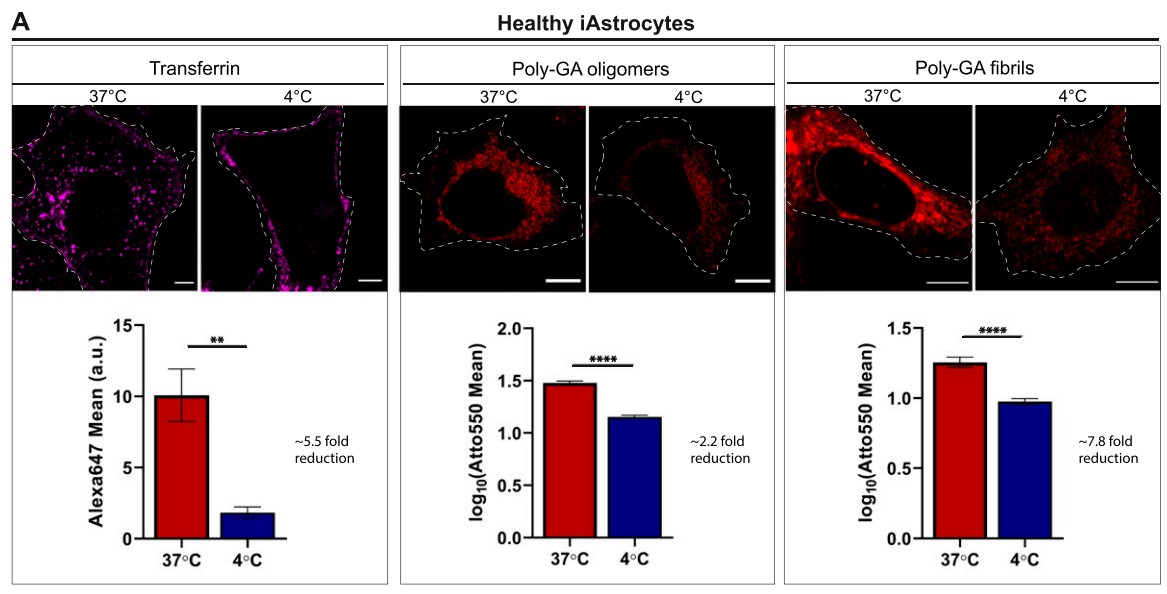

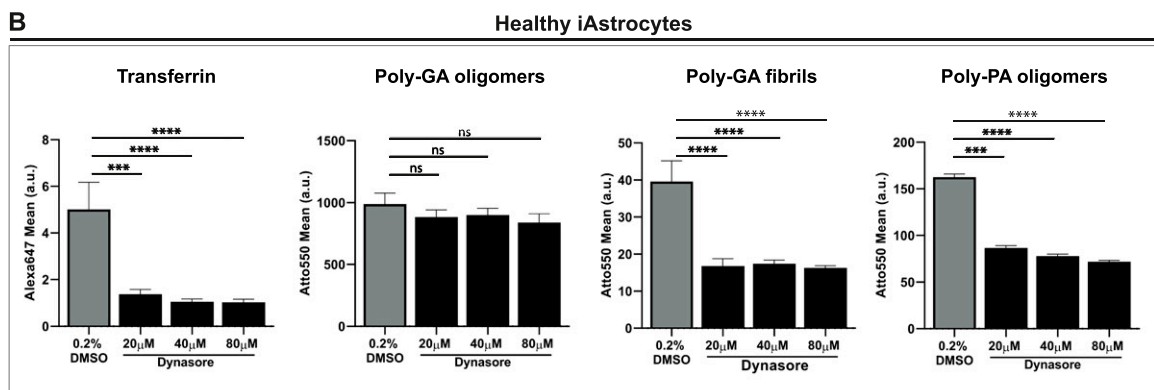

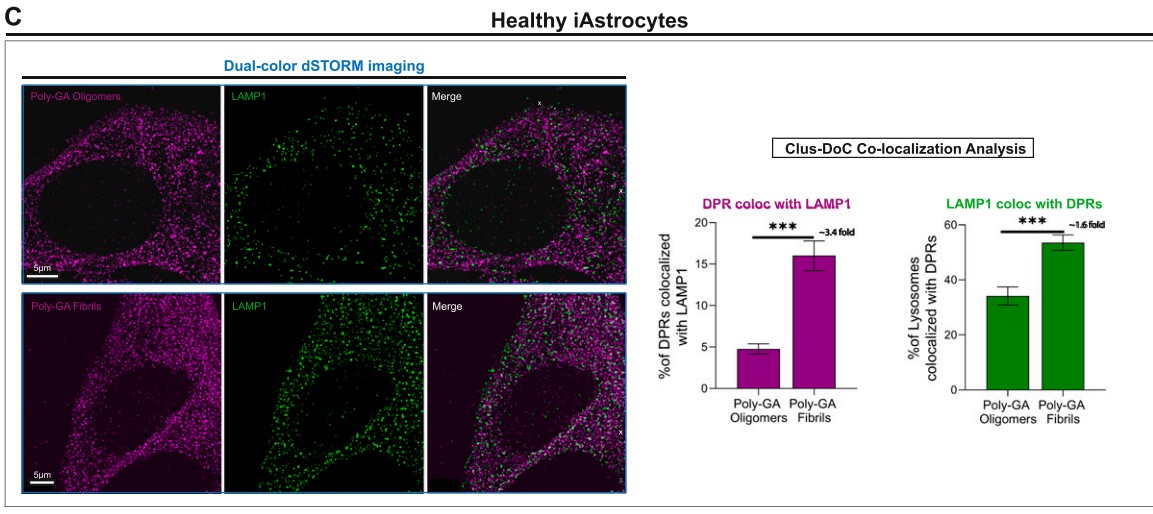

**Figure 4. Oligomeric vs fibrillar poly-GA dipeptide repeats entry-routes in glia.**
**(A)** Confocal images of Alexa647-Transferrin (control) and 1 μM ATTO550-Poly-GA aggregates after 1 h uptake at 37°C or at 4°C in healthy iAstrocytes. Quantification of the $\log_{10}$-transformed mean grey values is reported in bar graphs of mean ± SEM. ≥300 cells/condition. Kolmogorov–Smirnov non-parametric test after testing normal distribution with the Shapiro–Wilk test. **$P \leq 0.01$, and ****$P \leq 0.0001$. Scale bars = 5 μm. The data were collected from four independent biological replicates.
**(B)** Quantification of the Mean grey values of Alexa647-Transferrin (control) and 1 μM ATTO550-DPRs upon 1 h treatment with Dynasore (or 0.2% DMSO). Bar graphs of mean ± SEM. ≥250 cells/condition. Kruskal–Wallis non-parametric test with Dunn's multiple-comparisons after testing normal distribution with the Shapiro–Wilk test.

co-cultured with motor neurons, they are able to transmit DPRs to neuronal cells.

# Discussion

Pathological protein aggregates in neurons and glia are a hallmark of most neurodegenerative diseases (Eisele et al, 2015). In the context of *C9ORF72*-ALS/FTD, aggregation of the RAN translation-derived poly-GA DPRs is one of the proposed mechanisms for inducing proteasome impairment, DNA damage, cognitive disability, motor deficits, pro-inflammatory responses, and neurodegeneration, as shown by numerous cell culture and in vivo studies (Mori et al, 2013; Zhang et al, 2016; Schludi et al, 2017; Guo et al, 2018; LaClair et al, 2020; Nihei et al, 2020).

In our work, we began our analysis of poly-GA DPR spread by investigating the process of poly-GA oligomerization and fibril formation. We first showed that the in vitro coalescence of poly-GA and poly-PA oligomers increased in correlation with protein concentration and incubation length. However, whereas poly-GAs formed large and solid-like assemblies, poly-PAs produced small spherical liquid-like droplets. This result is in agreement with the reduced toxicity of poly-PA DPR species (Mizielinska et al, 2014; Vanneste et al, 2019) because maintenance of liquid-phase homeostasis was proposed to be non-pathogenic in protein aggregation (Patel et al, 2015; Peskett et al, 2018; Ray et al, 2020). Next, we demonstrated that poly-GA can form characteristic β-sheet amyloid fibrils in vitro upon 15-d incubation.

Because the transition to β-sheet fibrils exposes hydrophobic amino acid residues (Landreh et al, 2016), we speculate that poly-GA could lead to significant problems of insolubility as previously described in cell culture studies (Zhang et al, 2014; Lee et al, 2017; Ohki et al, 2017; LaClair et al, 2020; Nihei et al, 2020).

We then compared cellular uptake of poly-GA oligomers and fibrils to determine whether differences in the aggregation stage could lead to differences in cell entry. Interestingly, poly-GA oligomers are taken up more easily than poly-GA fibrils or other oligomeric species such as poly-PA. At least a fraction of poly-GA DPRs is internalised by endocytosis in iAstrocytes. Intriguingly, when more specific pathways of endocytosis were pharmacologically inhibited, such as those depending on the GTPase protein dynamin, only the uptake of poly-GA fibrils was reduced but no change was observed in poly-GA oligomers uptake. This result might be explained by an intrinsic capability of small oligomers to escape lysosomal surveillance. One of the proposed mechanisms for avoiding lysosomal degradation is inducing lysosomal damage, a feature that has been linked to the toxicity of certain oligomeric aggregates (Lee et al, 2008; Kandimalla et al, 2009; Tomic et al, 2009; Jiang et al, 2017); poly-GA oligomers indeed showed reduced colocalization with LAMP1-endolysosomes compared with poly-GA

fibrils. To investigate lysosomal damage in neurons, poly-GA oligomers or fibrils were incubated with mouse cortical neurons. When administered to the neuronal soma in oriented neuron cultures, these DPRs were taken up and transported along the axons, accumulating in stalling lysosomes found in axonal swellings. Importantly, the lysosomal population which colocalized with DPRs presented abnormalities in motility, speed and size compared with the non-colocalized counterpart. Because of the polarity of neurons, the regulation of lysosomal motility and size is especially important: lysosomes need to access specific cytoplasmic locations to perform their various functions (Pu et al, 2016), and lysosomal size is critical for fission and fusion events (de Araujo et al, 2020). After evaluating in situ lysosomal activity, cathepsin, and galectin-3 protein levels, we found no clear lysosomal damage mechanism following exposure to poly-GA. However, we noticed that poly-GA fibrils boosted transcript levels of the ATP6V0E1 gene, which encodes a multi-subunit ATPase component. As demonstrated in HeLa cells with a drop in lysosomal pH (LysoSensor) after poly-GA exposure, up-regulation of ATP6V0E1 mRNA may represent a biological response to poly-GA clearance, presumably by maintaining or enhancing lysosomal pH acidic gradients via V-ATPase activity. Our recombinant DPRs have the potential to disrupt important lysosomal partners such as microtubules and microtubule motors. Indeed, arginine-rich DPRs have been shown to impede the translocation of dynein and kinesin-1 motor complexes as well as bind microtubules, promoting their pausing and detachment (Fumagalli et al, 2021). We hypothesize that comparable processes are at work in the behaviour observed after poly-GA internalisation because our data reveal that poly-GA aggregates disseminate throughout the cell following nocodazole treatment, implying the presence of poly-GA–microtubule interactions.

We finally sought to investigate whether poly-GA oligomers and poly-GA fibrils could show differences in the ability to undergo cell-to-cell propagation, a process previously observed for various DPR species in neuronal cell cultures (Chang et al, 2016; Westergard et al, 2016; Zhou et al, 2017; Khosravi et al, 2020) as well as in the Drosophila nervous system (Morón-Oset et al, 2019). We proceeded to explore these questions by establishing a co-culture system between iAstrocytes and Hb9-GFP mouse motor neurons; this system was used because of two main reasons: (i) the largely unexplored role of glia in DPR propagation with relation to neurons, (ii) and the previously described role of astrocytes as "hubs" for intercellular propagation of protein aggregates (Loria et al, 2017; Victoria & Zurzolo, 2017; Wang et al, 2019). Our results showed that poly-GA DPRs undergo astrocyte-to-neuron propagation, with the fibrils being six-times more efficient in transferring to motor neurons compared with oligomers. Thus, DPRs at later stages of aggregation might be more prone to transfer from affected to naïve cells, which is in agreement with previous findings related to other amyloid proteins (Ray et al, 2020). Somewhat surprisingly, we were unable to

***P ≤ 0.001, and ****P ≤ 0.0001. The data were collected from three independent biological replicates. **(C)** Healthy iAstrocytes were imaged by dual-colour STORM after 24 h incubation with 0.5 µM ATTO647N-PolyGA oligomers or fibrils (magenta) and anti-LAMP1 staining (green). Clus-DoC colocalization analysis for Poly-GA relative to LAMP1 (magenta graph), and LAMP1 relative to Poly-GA (green graph) shows the respective %colocalized molecules (among total molecules detected). Bar graphs of mean ± SEM; graphs are indicative of 30 regions-of-interest (4 × 4 µm) per condition chosen only in artefact-free regions. Unpaired two-tailed *t* test with Welch's correction. ***P ≤ 0.001. Scale bar = 5 µm. The data were collected from three independent biological replicates.

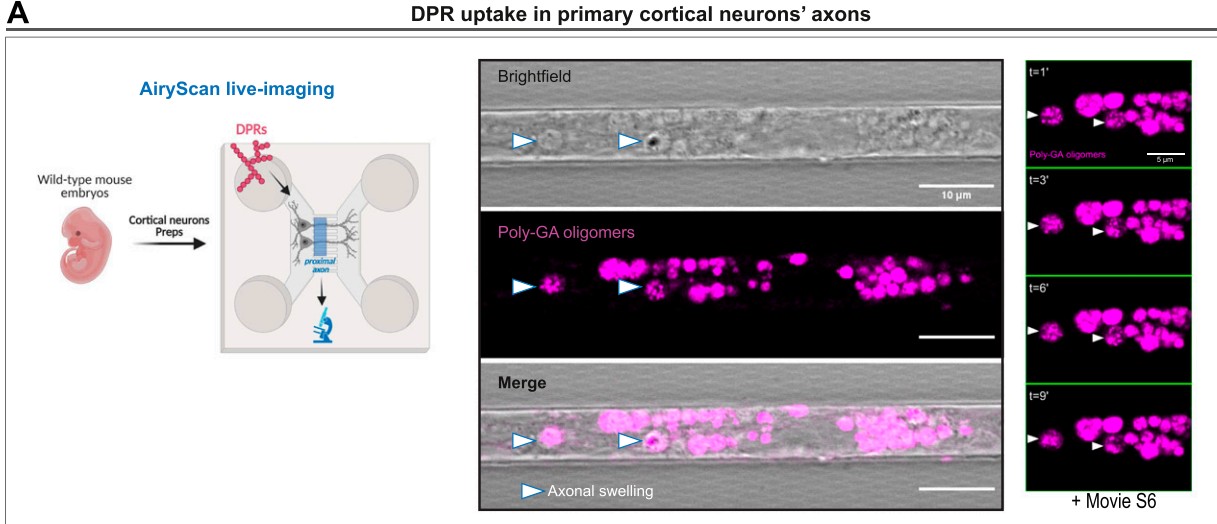

**A** DPR uptake in primary cortical neurons' axons

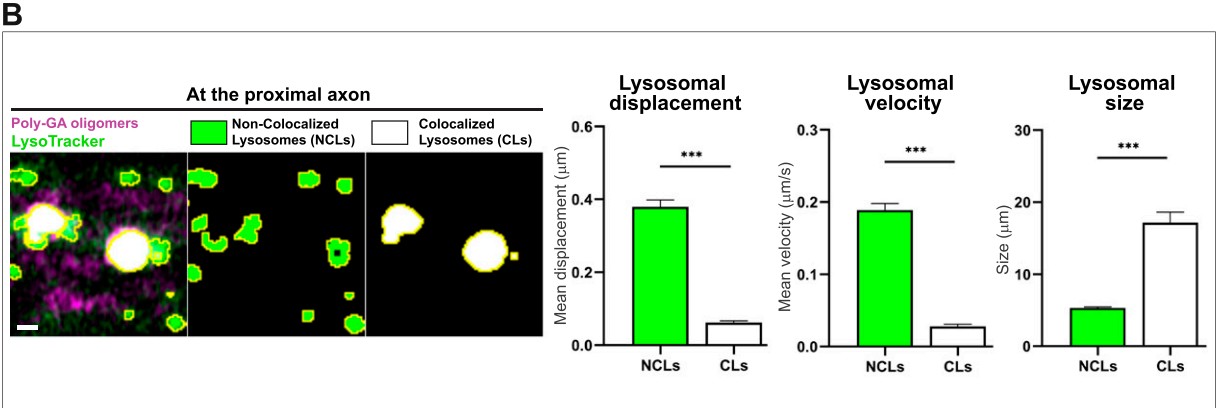

**B**

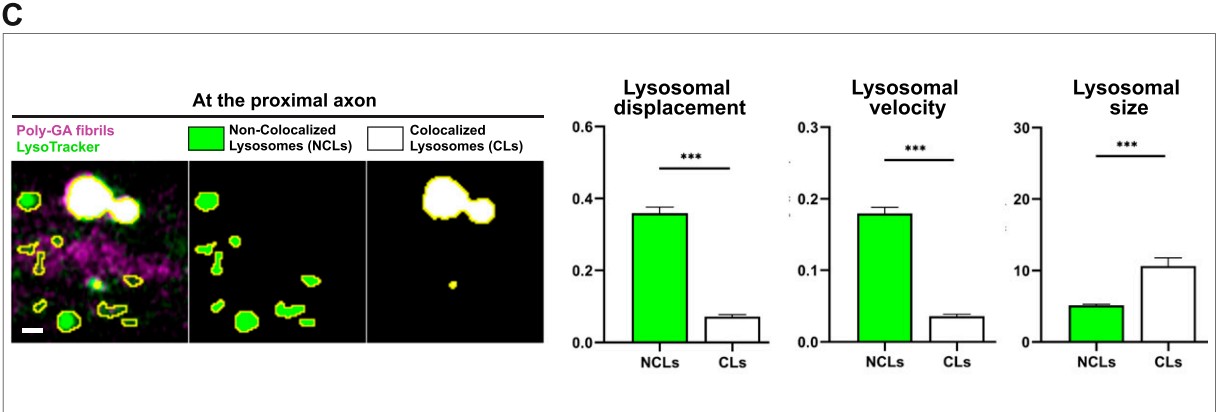

**C**

**Figure 5. Poly-GA aggregates colocalize with aberrantly enlarged endolysosomal organelles after uptake in neurons.**
**(A)** Confocal images showing accumulation of 1 µM poly-GA assemblies in large axonal swellings (arrow heads) along the axons of primary mouse cortical neurons. A combination of bright-field imaging and fluorescence detection of the ATTO550-labelled poly-GAs was used to detect these swellings specifically in the axons residing in the microfluidic chamber microgrooves (scale bar = 10 $\mu$m). By zooming into few axonal swellings with higher resolution (AiryScan mode), during live-imaging, we report the presence of small poly-GA assemblies' particles erratically moving within each axonal swelling overtime (right panel, scale bar = 5 $\mu$m); corresponding movie is shown in Video 6. **(B, C)** Colocalization analysis between all the poly-GA oligomers (B) or the poly-GA fibrils (C) and the lysosomes contained in the cortical neurons' proximal axons. By colour deconvolution we separated "non-colocalized lysosomes" (NCLs) and "colocalized lysosomes" (CLs) to analyse displacement, speed and size. Scale bars = 1 $\mu$m. Bar graphs of mean ± SEM. Unpaired two-tailed $t$ test with Welch's correction. ***$P \leq 0.001$. The data were collected from three independent biological replicates. **(A)** Figure on the left of panel (A) was created with BioRender.com under academic license.

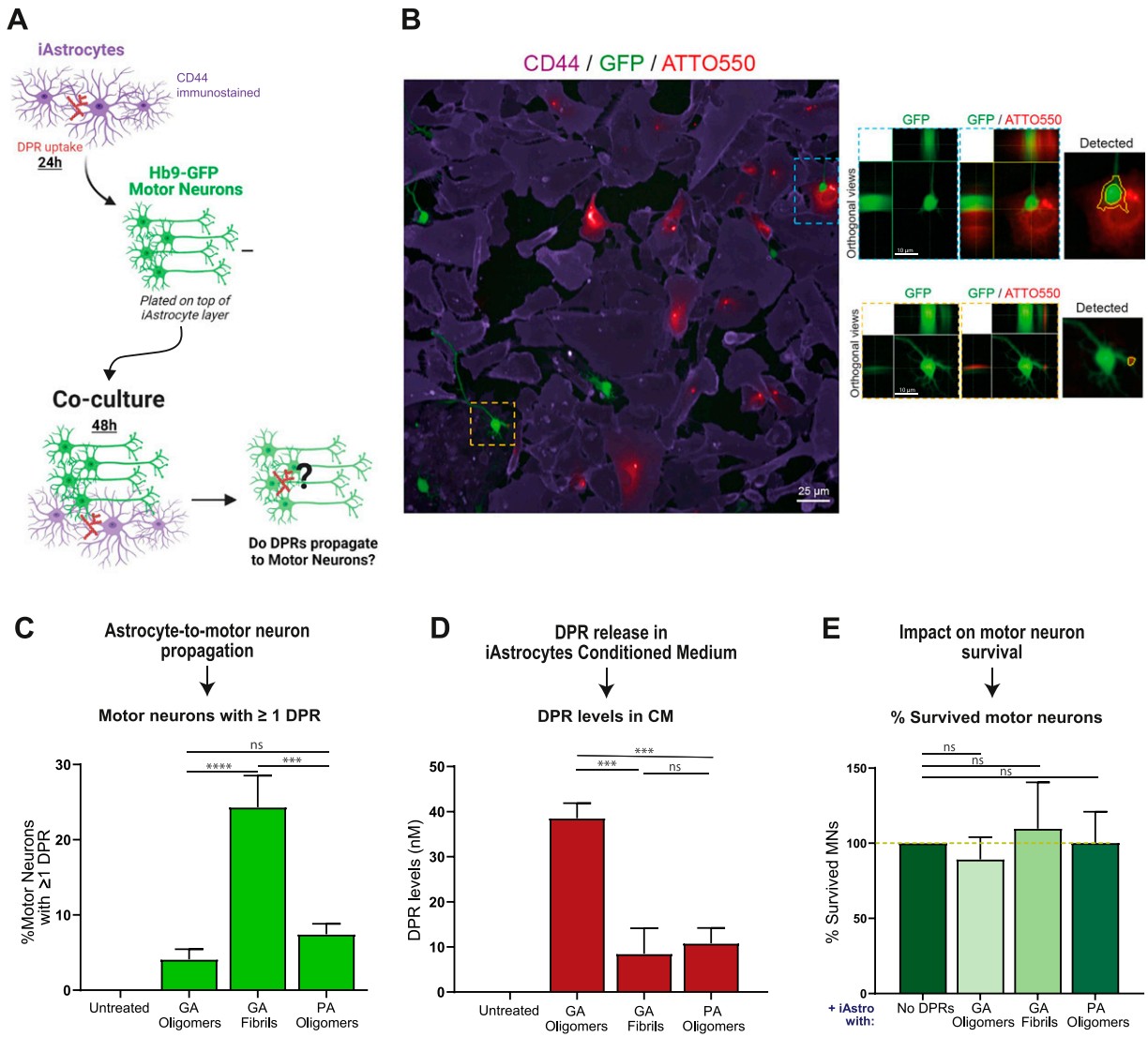

**Figure 6.  Alanine-rich dipeptide repeats (DPRs) undergo astrocyte-to-motor neuron propagation.**
**(A)** Schematic representation of the iAstrocytes-MNs co-culture system; the figure was created with BioRender.com under academic license. **(B)** Orthogonal views from confocal images show the presence of 1 μM ATTO550 DPRs (red) in the GFP-positive MNs (green); CD44 is shown in violet as the astrocytic marker. **(C)** Quantification of the percentage of motor neurons containing at least one DPR aggregate in the various 48 h co-culture systems. Bar graphs of mean ± SEM. One-way ANOVA with Tukey's multiple-comparisons test. ~120 neurons/condition. ns = $P > 0.05$, ***$P \leq 0.001$, and ****$P \leq 0.0001$. The data were collected from three independent biological replicates. **(D)** Quantification of DPR levels present in the conditioned medium of healthy iAstrocytes, via spectrophotometric analysis. Bar graphs of mean ± SEM. One-way ANOVA with Tukey's multiple-comparisons test. ns = $P > 0.05$, and ***$P \leq 0.001$. The data were collected from four independent biological replicates. **(E)** Quantification of motor neuron survival upon 48 h co-culture with iAstrocytes containing and transmitting DPRs. Bar graphs of mean ± SEM. One-way ANOVA with Tukey's multiple-comparisons test. ~120 neurons/condition. ns = $P > 0.05$. The data were collected from three independent biological replicates.

detect any cytotoxicity in co-cultured motor neurons associated with poly-GA DPRs, nor any increase in APOJ lipoprotein in iAstrocyte CM, which has previously been linked to astrocyte-induced toxicity (Guttenplan et al, 2021). Cytotoxicity might perhaps require the synergistic presence of other DPR species (such as poly-GR and poly-PR), which were not included in our experimental settings. Nonetheless, our data suggest an important role for astrocytes in the transmission of C9ORF72-derived DPRs to neighbouring motor neurons. Whether this mechanism could constitute a "non-cell autonomous" contributor in *C9ORF72*-ALS/FTD neuropathology remains to be fully elucidated. Although DPR *post-mortem*

inclusions have been detected to a lesser extent in glial cells than in neurons, thus receiving less attention, we speculate that DPR aggregation in glia may still go undetected earlier in disease and play a role in disease pathogenesis. For instance, astrocytes could act as hubs for the continuous internalisation and redistribution of DPRs. Interestingly, in cell culture studies, the ability of poly-GA aggregates to transmit from cell to cell has been proven to disseminate proteasome inhibition and TDP-43 disease in neighbouring cells (Khosravi et al, 2020). Although the capacity of poly-GA to propagate in vivo in mammals has yet to be determined, poly-GA derived-proteasome impairment could be upstream of TDP-43

disease and be intertwined with impaired autophagy function caused by *C9ORF72* haploinsufficiency. Such proteasome and autophagy dysfunctions may exacerbate vulnerability not only in neurons but also in astrocytes, making these glial cells less effective in clearing pathological aggregates and in supporting neuronal homeostasis; as shown in the context of ALS *FIG4* mutations (Ferguson et al, 2009) and in a lysosomal storage disease study (Di Malta et al, 2012).

It should be noted that findings in cell culture settings may not reflect the situation in animal models or humans: for example, DPR concentrations in vivo in patients may vary widely because of the endogenous pattern of C9ORF72 expression, and the short DPR length used in our study cannot reflect the potential influence of longer DPR repeats on our described cellular events. However, our study remains informative for providing some fundamental insights into the biology related to poly-GA aggregation, binding, uptake and cell-to-cell propagation in glia and neurons. Unless we improve our understanding of non-neuronal cell contributions and mechanisms preceding disease onset, the clinical variability in disease progression will continue to confound the design and evaluation of ALS and FTD treatments.

# Materials and Methods

## Cell culture

1321N1 astrocytoma cells were cultured in DMEM (Sigma-Aldrich) supplemented with 10% FBS (Gibco) and 5 U ml$^{-1}$ Penstrep (Lonza). Hb9-GFP mouse stem cells were cultured as described (Wichterle et al, 2002) and differentiated into motor neurons with 2 $\mu$M retinoic acid (Sigma-Aldrich) and 1 $\mu$M Smoothened Agonist (SAG) (Millipore) for 5 d. Embryoid bodies were then dissociated with papain. All cells were maintained in a 37°C incubator with 5% $CO_2$.

## Conversion of skin fibroblasts to iNPC

Skin fibroblasts from one healthy control (see Table 1) were reprogrammed as previously described (Meyer et al, 2014). Briefly, $10^4$ fibroblasts were grown in one well of a six-well plate. Day 1 post-seeding the cells were transduced with retroviral vectors containing Oct 3/4, Sox 2, Klf 4, and c-Myc. Following 1 d of recovery in fibroblast medium, DMEM (Gibco), and 10% FBS (Life Science Production), the cells were washed 1× with PBS, and the culture medium was changed to Neural Progenitor Cell (NPC) conversion medium comprising of DMEM/F12 (1:1) GlutaMax (Gibco), 1% N2 (Gibco), 1% B27 (Gibco), 20 ng/ml FGF2 (PeproTech), 20 ng/ml EGF (PeproTech), and 5 ng/ml heparin (Sigma-Aldrich). As the cell morphology changes and cells develop a sphere-like form, they can be expanded into individual wells of a six-well plate. Once an iNPC culture is established, the medium is switched to NPC proliferation medium consisting of DMEM/F12 (1:1) GlutaMax, 1% N2, 1% B27, and 40 ng/ml FGF2.

## iAstrocyte differentiation and co-culture system

iAstrocytes were yielded as previously described (Meyer et al, 2014; Hautbergue et al, 2017). Briefly, iNPCs were switched to astrocyte proliferation medium, which includes DMEM (Thermo Fisher Scientific), 10% FBS (Life science production) and 0.2% N2 (Gibco). Cells were grown in 10 cm dishes coated with fibronectin for 7 d unless otherwise stated. For the co-culture system, we treated iAstrocytes with 1 $\mu$M DPRs for 24 h, then plated Hb9-GFP mouse motor neurons on top of the astrocyte layer and kept this co-culture system for 48 h before fixation and confocal imaging.

## Primary mouse cortical neurons

Primary cortical neurons were produced from E15.5 embryos of wild-type C57BL/6 mice. Brains were harvested and hemispheres were divided. In HBSS−/− medium, meninges and midbrain were removed to isolate the cortical tissue, which was then incubated with trypsin (Gibco) for cell dissociation. Single-cell suspension was obtained by mechanical pipetting in appropriate trituration solution (HBSS+/+ with 1% Albumax, 25 mg trypsin inhibitor, 10 mg/ml DNAse stock). Finally, cortical neurons were resuspended in Neurobasal medium (Thermo Fisher Scientific) with B27 (Gibco), 1% Pen/Strep (Thermo Fisher Scientific), and 1% glutamine (Lonza) and seeded on Xona silicon device (#RD450) coupled with a 35-mm dish previously coated with poly-D-lysine (Sigma-Aldrich). Cells were maintained in a 37°C incubator with 5% $CO_2$, changing the medium every 2 d. Neurotrophic factors (2 ng/ml Brain-derived neurotrophic factor [BDNF], 2 ng/ml Glial cell line-derived neurotrophic factor [GDNF]) were added into the medium to favour the correct directionality of axonal growth through the microgrooves. After 12 d in culture, the cells were stained with LysoTracker Green (Thermo Fisher Scientific) and live-imaging was performed with Airyscan microscopy (LSM 880; Zeiss) at 1.5 Hz for 2 min (561 and 488 channels; 63× 1.4 NA oil immersion lens).

## Dipeptide repeat proteins cloning, purification, and labelling

The V5-tagged 34-GA repeat was obtained using an expandable cloning strategy with Age1 and Mre1 as compatible enzymes (Mcintyre et al, 2008). A "start acceptor" pCi-Neo vector (Promega) was first constructed by cloning a V5-3xGly/Ala insert into the Xho1/Not1 sites (ctc gag gcc acc atg ggc aaa ccg att ccg aac ccg ctg ctg ggc ctg ctg gat agc acc ggt gca ggt gct ggc gcc ggc gga tcc gaa ttc tag ccg cgg ccg c) and a "start donor" vector with a 14xGly/Ala insert (ctc gag acc ggt gca ggt gct gga gct ggt gca ggt gct gga gca ggt gca ggt gct gga gct ggt gca ggt gct gga gca ggt gct ggc gcc ggc gga tcc gaa ttc ccg cgg ccg c) in the Xho1/Not1 sites. These vectors ("start acceptor" and "start donor") were then used to propagate the GA repeats to construct 34-GA repeats. The plasmid pAG416-Gal, which encodes for the PAx50 dipeptide repeats, was a gift from Aaron Gitler (plasmid #84902; Addgene; http://n2t.net/addgene:84902) (Jovičić et al, 2015). DNA sequences encoding V5-tag followed by 34 repeats of GA or encoding FLAG-tag followed by 50 repeats of PA were subcloned in a bacterial expression vector containing an N-terminal 6xHis-Tag and a TEV protease cleavage site (pETM-11 vector, EMBL). The pETM-11 vector was a gift from Frank Schulz (plasmid #108943; Addgene; http://n2t.net/addgene:108943) (Dirkmann et al, 2018). In particular, the V5-tagged 34-GA construct was subcloned in the pETM-11 vector using NcoI/NotI restriction sites; whereas the FLAG-tagged 50-PA repeat construct was subcloned in the pETM-11

**Table 1.  List and characteristics of control-derived cells used in this study.**

| Patient sample | Cell type | Gender | Ethnicity | Age at biopsy collection (years) |
|---|---|---|---|---|
| 155v2 | Healthy control | Male | Caucasian | 40 |
| 161 | Healthy control | Male | Caucasian | 31 |
| CS14 | Healthy control | Female | Caucasian | 52 |

vector using NcoI/XhoI sites. An additional pCI-Neo vector with DNA sequences encoding V5-tag followed by 24 repeats of GP was subcloned in the pETM-11 vector using NcoI/NotI restriction sites. Transformation of bacterial cells (*E. coli* BL21) with plasmids containing these repeat constructs (V5-tagged 34-GA; FLAG-tagged 50-PA; V5-tagged 24-GP) was performed. Thus, proteins were expressed in *E. coli* BL21 and purified on a 5-ml Talon column (Clontech) loaded with cobalt. The proteins were eluted with a linear gradient of 12 ml from buffer A (20 mM Tris, pH 7.5, 250 mN NaCl, 5 mM imidazole, 1 mM $\beta$-mercaptoethanol, and glycerol 10% PMSF 0.1 mM) to buffer B (20 mM Tris, pH 7.5, 250 mM NaCl, 250 mM imidazole, 1 mM $\beta$-mercaptoethanol, and glycerol 10% PMSF 0.1 mM). Eluted fractions were analysed by SDS–PAGE, and proteins were quantified spectrophotometrically using a molar extinction coefficient of $\varepsilon_{GA}$ = 2,980 M$^{-1}$ cm$^{-1}$ and $\varepsilon_{PA}$ = 4,470 M$^{-1}$ cm$^{-1}$. Proteins were assembled into fibrils at 4°C without shaking for ≤14 d. Proteins were labelled with Atto-550 (Atto:DPR 5:1) or Atto-647N (Atto:DPR 2:1) dyes (#AD550-35, #AD647-35; Atto-Tec). Unreacted NHS-dye and non-fibrillar polypeptide was removed by centrifugation (100,000*g*, 30 min, 4°C). Labelled assemblies were fragmented by sonication for 5 min in 2-ml Eppendorf tubes in a Vial Tweeter powered by an ultrasonic processor UIS250v (250 W, 2.4 kHz; Hielscher Ultrasonic) to generate fibrillar particles that are suitable for endocytosis (with an average size of 45–55 nm), flash-frozen in liquid nitrogen and stored at −80°C. Immunoreactivity and fluorescent labelling of the generated GA/PA-repeat recombinant proteins were confirmed by resolving the samples with dot-blotting or protein gel electrophoresis (Fig S2).

The morphology of DPR assemblies was assessed by Transmission Electron Microscopy in a Jeol 1400 microscope before and after fragmentation following adsorption onto carbon-coated 200 mesh grids and negative staining with 1% uranyl acetate. The images were recorded with a Gatan Orius CCD camera (Gatan). The $\beta$-sheet amyloid component of fibrillar poly-GA assemblies was assessed and confirmed by Fourier-transform infrared spectroscopy as described (Brasseur et al, 2020). Importantly, before the addition to the medium, fibrillar DPRs were sonicated for 5 min at 80% amplitude with a pulse cycle of 5 s on and 2 s off (MSE Soniprep 150); this procedure is required to disperse the aggregated $\beta$-sheet assemblies.

### Coalescence measurements

For phase-separation experiments, soluble GA and PA DPRs (stock concentration = 100 $\mu$M) were diluted in ddH$_2$O to either 20, 10 or 1 $\mu$M. After vortexing, the mixture was pipetted onto glass-bottom slides (Ibidi), and assembly formation was monitored over time. Images were acquired immediately, as well as after 2 and 24 h using a Leica SP5 confocal microscope with a 63× 1.4 NA oil immersion objective, 561 channel. To capture fine details for 3D rendering, assemblies were imaged with a z-stack following Nyquist sampling for optimized pixel density. Huygens Professional version 19.10 (Scientific Volume Imaging, http://svi.nl) was used to deconvolve z-stack data using the CMLE algorithm (with SNR:10 and 40 iterations) and subsequently for 3D-volume and surface rendering thus generating Videos 1 and 2. The 3D rendering for Videos 3 and 4 was performed using the software Imaris v7.7.2 (Bitplane); no deconvolution was applied here. Number, circularity and size of DPR assemblies were quantified with FIJI (Schindelin et al, 2012) and plotted using GraphPad Prism 8. The FIJI plug-in Trainable Weka Segmentation (Arganda-Carreras et al, 2017) was used to finely measure the circularity of liquid droplets in heterogeneous PA assemblies (20 $\mu$M, 24 h) (Fig S3D and E).

### MSD($\Delta t$) analysis

Human iAstrocytes were exposed to 1 $\mu$M ATTO550-labelled Poly-GA DPR fibrils for 24 h. Cells were then washed several times with PBS and, while keeping them in 5% CO$_2$ and 37°C, z-stack live-imaging was performed by taking 60 frames at ~1 frame/s rate (1 frame = 1 full z-stack) at ~190 nm lateral resolution (Zeiss LSM880, airyscan mode). Single DPR aggregates were then detected as single particles and analysed by the open-source FIJI-plug-in TrackMate (Tinevez et al, 2017) using difference of Gaussians (DoG) detection (*estimated blob diameter* = 0.8 $\mu$m; *threshold* = 100) and Simple LAP tracker (*linking max distance* = 1 $\mu$m; *gap-closing* = 1 $\mu$m; *max frame gap* = 5). Video 5 was produced in 3D-volume rendering mode using Arivis Vision 4D software (Arivis AG) after complete image stack deconvolution was performed for each frame with Huygens Professional version 19.10 (CMLE algorithm, SNR:20, 40 iterations). The *xy* data for each tracked object were analysed to determine mean-squared displacements (*MSD*) as a function of time-step, $\Delta t$. For untethered non-interacting objects moving freely within the medium, $MSD(\Delta t)$ is expected to evolve linearly with a gradient equal to 4D, where *D* is the Brownian translational diffusion coefficient. If the object also experiences ballistic motion (i.e., constant translational velocity magnitude and direction), which could approximately describe microtubule transport (van den Heuvel et al, 2007), the $MSD(\Delta t)$ becomes quadratic and is represented by the equation: $MSD = 4D\Delta t + v^2\Delta t^2$, where *v* is the average ballistic velocity (Dunderdale et al, 2012). Consequently, $MSD(\Delta t)$ data were fitted to this expression to identify any trajectories that showed non-zero values for *v*, and so possibly reflect microtubule directed motion. For this analysis, the time range fitted was 20 s, and only trajectories greater than 30 s in length were analysed (380 separate trajectories met this criterion).

For the $MSD(\Delta t)$ analysis on DPR trajectories in primary mouse cortical neurons (Fig S6A and B), we first used TrackMate for

producing DPR tracks and then we implemented the MATLAB class @msdanalyzer written by Jean-Yves Tinevez (https://github.com/tinevez/msdanalyzer GitHub), already used in a previous study (Tarantino et al, 2014) and explained in its details (Miura & Sladoje, 2020). MSD plots and Log–Log fit plots were produced with MATLAB R2018b using the aforementioned class. A detailed explanation of how the analysis was performed (from the generation of TrackMate DPR tracks to the production of graphs after MSD analysis) can be found on GitHub at the following link: https://github.com/paoloM1990/Guide-to-TrackMate-Matlab-MSD.

### Flow cytometry

1321N1 human astrocytoma cells ($1.2 \times 10^6$/sample) were washed six times in PBS (to eliminate any remaining DPRs in the media), trypsinized, and then resuspended in 500 $\mu$l of PBS. Suspended cells were then analysed by using an LSRII flow cytometer (BD Bioscience) with excitation at 488 nm and BD FACSDiva software (version 8.0.1; BD Bioscience) to excite the ATTO550 fluorophore. Detection of ATTO550+ signal was set at 610/20 voltage. Cells not treated with DPRs acted as control producing the gating to discriminate between the ATTO550+ and the ATTO550− cells.

### Immunocytochemistry

After the addition of the DPRs, all the single-cell cultures were washed five to six times with PBS and fixed with 4% PFA for 30 min at room temperature. After fixation, cells were washed two times with PBS, permeabilized for 10 min with 0.1% Triton X-100:PBS and additionally washed twice with PBS. Subsequently, cells were incubated with the blocking agent 3% BSA for 30 min and then incubated overnight at 4°C with primary antibodies (in 3% BSA). Cells were then washed 3× with PBS and incubated for 1 h with the corresponding Alexa Fluor secondary antibodies (Thermo Fisher Scientific) at 1:1,000 dilution (in 3% BSA) and with Hoechst when needed. Cells were washed 3× with PBS, and coverslips were mounted onto glass slides using Fluoromount aqueous mounting medium (Sigma-Aldrich). In the case of imaging with Opera Phenix high-throughput system (PerkinElmer), fixed cells were imaged directly in PBS on a 96-well optical-bottom plate (#165305; Thermo Fisher Scientific), thus coverslipping was not required.

Primary antibodies used were: chicken anti-imentin (1:4,000, #AB5733; Millipore), mouse anti-$\alpha$-tubulin (1:1,000, #T9026; Sigma-Aldrich), rabbit anti-CD44 (1:1,000, #ab157107; Abcam), mouse anti-Cathepsin B (1:500, #sc-365558; Santa Cruz Biotechnology), mouse anti-Cathepsin D (1:1,000, #sc-377299; Santa Cruz Biotechnology), mouse anti-Cathepsin L (1:1,000, #sc-390367; Santa Cruz Biotechnology), mouse anti-Galectin 3 (1:100, #sc-374253; Santa Cruz Biotechnology) and mouse anti-LAMP1 (1:25, #ab25630; Abcam).

### Immunoblotting

To collect protein extracts, iAstrocytes were washed six times in PBS (to eliminate any remaining DPRs in the medium) and lysed in RIPA buffer (20 mM Tris–HCl, pH 7.5, 137 mM NaCl, 10% glycerol, 1% Triton X-100, 0.5% sodium deoxycholate, 2 mM EDTA, and 0.1% SDS, supplemented with protease inhibitor cocktail; Sigma-Aldrich) on ice

for 20 min. The protein extracts were then collected in the supernatant and the concentration of each protein extract was estimated using a BCA assay (Pierce). Equal quantities of protein were mixed with 4× loading buffer (0.4 M sodium phosphate, pH 7.5, 8% SDS, 40% glycerol, 10% 2-mercaptoethanol, and 0.05% bromophenol blue), heated to 95°C for 5 min, and processed for either dot-blot or Western blot. Nitrocellulose membranes (0.22 $\mu$m pores) were blocked in 1× TBS with 0.05% Tween (1× TBST) with 5% wt/vol non-fat dry milk for 1 h, and then incubated with primary antibodies in 1× TBST 5% wt/vol non-fat dry milk at either room temperature for 2 h or 4°C overnight. Primary antibodies used were: anti-V5 (1:1,000, #13202S; Cell Signaling Technology), anti-$\alpha$-tubulin (1:3,000, #T9026; Sigma-Aldrich), anti-GA repeat (1:1,000, #24492-1-AP; ProteinTech), anti-AP repeat (1:1,000, #24493-1-AP; ProteinTech). Membranes were then washed three times for 5 min with 1× TBST and incubated with either an anti-mouse IgG-HRP–conjugate (1:5,000, Cat. no. 172-1011; Bio-Rad) or an anti-rabbit IgG-HRP-conjugate (1:5,000, Cat. no. 12-348; Millipore). ECL substrate was then added to the membrane to enable detection, and non-saturated images were acquired using a G:BOX EF machine (Syngene) and GeneSys software (Syngene).

### Perturbation of endocytosis

Exposure of cells to low-temperature conditions is a commonly used method for nonspecific inhibition of endocytosis. Healthy control iAstrocytes were primed with 30 min of exposure to 4°C, and then the DPR assemblies (diluted in ice-cold DMEM) were delivered to the cells and incubated at 4°C for an additional time of 1 h. The same procedure was applied for Alexa647-labelled transferrin, an established marker of clathrin-coated pit endocytosis (Ehrlich et al, 2004). In parallel conditions, cells exposed to DPR assemblies or transferrin were incubated with DMEM at 37°C for 1 h. For the dynasore experiment, following the established protocol (Kirchhausen et al, 2008), cells were primed for 30 min with Dynasore (or 0.2% DMSO) and incubated at 37°C in serum-free medium before addition of transferrin or poly-GA fibrils for 1 h. Cells were subsequently fixed in Glyoxal solution (pH = 5) for 30 min (Richter et al, 2018) and imaged with confocal microscopy (LSM 880; Zeiss, Airyscan mode) from the plane of sharp focus. From the images, using FIJI and creating a macro (https://github.com/paoloM1990/Quantification-of-cell-fluorescent-intensity), the *Mean grey values* of Atto550 or Alexa647 whole-cell signals were calculated. In the case of the 37–4°C endocytosis experiment, *mean grey values* were log-transformed ($\log_{10}$) only for better graph visualization.

### Lysosomal assays

Lysosomal in situ enzyme activity was measured by using lysosome-specific self-quenched substrate (Cat. no. ab234622; Abcam) at manufacturers recommended dosage. In brief, primary mouse cortical neurons were exposed to poly-GA DPRs for 24 h (or were left untreated), and lysosome-specific self-quenched substrate was added during the final 1 h of the 24-h period. Cells were then fixed with 4% PFA at room temperature for 15 min before being imaged with a Zeiss LSM 880 confocal microscope on glass-bottom slides

(Ibidi). FIJI was used to analyse the images, and the mean fluorescence intensity of the substrate was quantified per neuronal cell.

Lysosomal staining with the fluorescent acidotropic probe, LysoSensor Green DND-189, was performed according to the manufacturer's recommendations (#L7535; Thermo Fisher Scientific). Briefly, HeLa cells on a 96-Well Optical-Bottom Plate (#165305; Thermo Fisher Scientific) were exposed to poly-GA DPRs for 24 h (or were left untreated), and LysoSensor Green was added during the final 1 h of the 24-h period (1 $\mu$M final concentration). After incubation, live cells were transferred to the PerkinElmer Opera Phenix high-throughput system for imaging (40× 1.1 NA lens). Using the Columbus Image Analysis System, the mean fluorescence intensity of LysoSensor Green was quantified per cell (PerkinElmer).

### mRNA isolation and quantitative real time PCR

Primary cortical neurons were produced from E15.5 embryos of wild-type C57BL/6 mice. Total RNA from primary mouse cortical neurons (after 10 d in culture) exposed to poly-GA oligomers or poly-GA fibrils (or untreated) for 24 h was isolated using RNeasy Mini Kit (Cat. no. 74104; QIAGEN), according to the manufacturer's manual. During RNA extraction, treatment with DNase I was applied to get rid of contaminating DNA. RT-qPCR was carried out using the QuantiFast SYBR Green RT-PCR Kit (QIAGEN). Briefly, a 10-$\mu$l volume reaction was set up by using 2 $\mu$l total RNA (diluted to a concentration of 10 ng/$\mu$l in nuclease-free H$_2$O), 5 $\mu$l 2× QuantiFast SYBR Green RT-PCR Master Mix, 1 $\mu$M forward primer, 1 $\mu$M reverse primer, 0.1 $\mu$l QuantiFast RT mix and nuclease-free H$_2$O. Following an initial reverse transcription step at 50°C for 10 min and a 5 min denaturation step at 95°C, the cDNA was amplified by 39 cycles of 95°C for 10 s followed by a combined annealing/extension step at 60°C for 30 s, and subsequent melt curve analysis (to ensure primer specificity), with data collected over a temperature range of 65–95°C in 0.5°C increments. RT-qPCR was performed on a Bio-Rad C1000 Touch Thermal Cycler. Bio-Rad CFX Manager software was used to analyse signal intensity and relative gene expression values were determined using the ΔΔCt method, with GAPDH RNA used as a reference gene. The RT-qPCR product was then visualized on a 2% agarose gel, after loading 10 $\mu$l of product along with 2 $\mu$l of 6× gel loading dye. After electrophoresis (at 120 V for 40 min), the gel visualization showed RT-qPCR products matching the expected bp size (data not shown). For the primer sequences used in this study, see Table 2.

### Colour deconvolution for differentiating two lysosomal populations

We developed custom scripts in MATLAB to produce colour deconvolution algorithms that separated "non-colocalized lysosomes" (NCLs) and "colocalized lysosomes" (CLs). NCLs corresponded to the signal of LysoTracker Green devoid of any overlapping with ATTO550-DPRs signal. CLs, instead, corresponded to the merged signal generated by LysoTracker Green–ATTO550-DPR colocalization. Briefly, eight-bit RGB time-lapse images were split into individual frames and a fuzzy colour detection algorithm was applied to identify and isolate regions of interest. Resulting images were binarised by applying a global Otsu threshold, and then de-noised using a median filter.

### Release of DPRs in the CM

To detect the cell release of ATTO550 DPRs into the CM, we have initially added 1 $\mu$M of DPRs to the culture medium for 24 h. After DPR uptake, cells were washed at least five times with PBS to remove remaining assemblies in the medium and then incubated for 24 h with PhenolRed-free FBS-free DMEM. This CM was harvested in tubes which were subsequently centrifuged at 200$g$ for 4 min to remove any remaining debris and dead cells. Finally, the HTS microplate reader PHERAstar FSX (BMG LABTECH) was used to measure ATTO550 fluorescence intensity (thus relative DPR concentration) in the CM on a 96-well plate. The optical module used for the fluorescence intensity measurement of each well was set to 540–20 nm (excitation light) and 590–20 nm (emission light), covering the whole area of the well. Importantly, we tested for linear dependence of fluorescence on concentration to evaluate potential inner filter effect (Fonin et al, 2014) by using nine known dilutions of purified DPRs in PhenolRed-free FBS-free DMEM; the plotted fluorescence values originated a standard curve with R$^2$ > 0.95.

### ELISA

APOJ (also known as clusterin) was measured in CM from either untreated or treated iAstrocytes for 24 h with various recombinant DPRs (poly-GA$_{34}$ fibrils, poly-GA$_{34}$ oligomers, poly-PA$_{50}$ oligomers, and poly-GP$_{24}$ oligomers). The CM was collected in tubes and centrifuged at 500$g$ for 4 min to remove any remaining debris and dead cells. After that, samples were diluted 1:10 and analysed within the range of the standard curve. APOJ protein levels were measured using a human APOJ ELISA kit (#KE00110; ProteinTech) following the manufacturer's instructions. The HTS microplate reader PHERAstar FSX (BMG LABTECH) was used to measure the absorbance of APOJ standards at 450 nm with the correction wavelength set to 630 nm. Regression analysis using the Four-parameter logistic curve-fit (4-PL) method was used to determine the best-fit standard curve (Fig S8A).

### dSTORM imaging

#### *Sample preparation*
High-precision (#1.5H Thickness) glass coverslips (CG15CH2; ThorLabs) were thoroughly rinsed in deionized water and dried. The glass coverslips were coated for 5 min at room temperature with fibronectin diluted in PBS (1:400) before iAstrocytes were plated.

#### *Immunocytochemistry and staining for dSTORM imaging*
Healthy control iAstrocytes were exposed to 0.5 $\mu$M ATTO-647N-labelled DPR fibrils and oligomers for 24 h. Cells were then washed six times with PBS to remove the remaining DPRs in the medium and then fixed for 60 min in 4% PFA (+0.2% glutaraldehyde) diluted in PBS. This long fixation period was used to minimize molecule motility (Tanaka et al, 2010).

For dual-colour dSTORM imaging, PFA-fixed cells were quenched in 50 mM NH$_4$Cl in PBS for 5 min at RT, followed by permeabilization

**Table 2. DNA sequences of RT-qPCR primers (*Mus musculus*) as they were ordered from Sigma-Aldrich.**

| Gene symbol | Primer forward (5′-3′) | Primer reverse (5′-3′) |
|---|---|---|
| ATP6V0E1 | GGGTCCTAACCGGGGAGTTA | ACAGAGGATTGAGCTGTGCC |
| CTSB | GCTCTTGTTGGGCATTTGGG | ACTCGGCCATTGGTGTGAAT |
| CTSD | CTTGGGCATGGGCTACCCTC | TTGCCCTTCTGGGTCCCTGTT |
| CTSL | CGCCTTCGGTGACATGACCA | TCTTGTGCTTCTGGTGGCGG |
| GAPDH | GGTCATGAGCCCTTCCACAA | TGAAGGGTGGAGCCAAAAG |
| GBA | GGGCAGCAAACTCCCTAGCAG | GGATGCAGGGTTGGGCACCATA |
| MCOLN1 | TGCTGTGGACCAGTACCTGA | GTAGTACCGCTGGCAGAGAG |

with 0.1% Triton X-100 in PBS and blocking of non-specific binding sites with 2% BSA. Staining with anti-LAMP1 Mouse primary mAb (1:25, Cat. no. ab25630; Abcam) was performed overnight at 4°C. Cells were washed with 3× PBS for 5 min and incubated for 2 h with anti-mouse IgG F(ab) ATTO488 (H + L) (Cat. no. 2112-250UG; HyperMOL) to yield minimal linkage error (Hust et al, 2007). Post-fixation in 4% PFA (+0.2% glutaraldehyde) for 30 min was applied to further reduce molecule motility when needed. Coordinates were tracked with Nikon NIS-Element software for computational drift correction and, in a separate sample, tetraspeck beads (100 nm diameter; Invitrogen) were imaged as fiduciary landmarks for chromatic re-alignment. Dual-colour dSTORM imaging was performed under reducing condition with Tris buffer 50 mM with 10 mM NaCl (pH 8), glucose (10%), cysteamine (1 M), glucose oxidase (5 mg/ml), and catalase (4 mg/ml). Approximately 8,000–10,000 frames per channel were acquired. Diffraction limited images of each channel were also acquired, providing the reference for subsequent NanoJ-SQUIRREL analysis (Culley et al, 2018) of image artefacts (Fig S5C).

All imaging was carried out on an inverted Nikon Eclipse Ti microscope equipped with a 100× oil immersion objective (1.49 NA) using an Andor iXon EMCCD camera (image pixel size, 151.57 nm). ATTO-647N and ATTO-488 were imaged using 639 and 488 nm lasers for a 10- or 20-ms exposure time. We used Nikon NIS-Elements software for both image acquisition and reconstruction. After image reconstruction, the package ChriSTORM (Leterrier et al, 2015) was used for translating NIS-Elements localization files into compatible files for image rendering by the open-source FIJI plugin ThunderSTORM (Ovesný et al, 2014). Thus, ThunderSTORM enabled the ultimate visualization of data acquired by STORM imaging (Fig S5A). Using the Nikon NIS-Elements software, single molecules were localised with a lateral localization accuracy of ~20 nm for 647 channel and of ~45 nm for 488 channel based on the Thompson equation (Thompson et al, 2002) (Fig S5B). In addition, by using NanoJ-SQUIRREL FIJI plug-in, we have implemented block-wise FRC resolution mapping to provide local resolution measurements of our dSTORM dataset (Culley et al, 2018) (Fig S5B).

### Colocalization analysis in dual colour dSTORM

For colocalization analysis, graphs are indicative of ~5–6 healthy iAstrocytes with five ROIs taken in regions juxtaposed to the nucleus of each cell (n = 3 biological experiments). The choice of ROIs was based by excluding artefact-rich areas with the FIJI plug-in NanoJ-Squirrel (Culley et al, 2018). For colocalization analysis, we used the open-source software Clus-DoC (Pageon et al, 2016) to generate colocalization maps (Co-Loc maps) which highlighted areas of molecular interaction between the two channels with the following parameters: L(r)-r radius = 20 nm; Rmax = 500 nm; Step = 10 nm; Colocalization threshold = 0.4; Min colocalized points/cluster = 10.

### 3D dSTORM

Nikon NIS-Element software and a Nikon Eclipse Ti microscope were used for 3D STORM imaging. We used an astigmatic lens in the light path—directly in front of the camera. To summarise, the NIS-Element calibration algorithm was first applied to tetraspeck beads in z-stack mode (600–800 nm of z-stack). Because of the astigmatism, the point spread function of the beads changes along the z-axis; this point spread function change is used by the calibration algorithm to estimate the position along the z-axis. For each imaged channel, a new calibration was applied (although in our case we only imaged ATTO647N dye, thus one channel).

### Quantification and statistical analysis

All data are presented as means ± SEM or means ± SD, where indicated. On normally distributed data, statistical differences were analysed using unpaired two-tailed *t* test (with Welch's correction, when SDs were not equal) for pairwise comparisons or one-way ANOVA (with Tukey's correction) for comparing groups of more than two. On non-normally distributed data, the non-parametric Kolmogorov–Smirnov test or Kruskal–Wallis test (with Dunn's multiple comparisons) were used for pairwise comparisons or for comparing multiple groups, respectively. Normal distribution was tested with Shapiro–Wilk test and Q–Q plot. $P < 0.05$ was considered statistically significant. All graphs and tests were generated using GraphPad Prism 8.

### Ethics statement

Ethical approval to use iPSCs or iNPCs is in place for this project (REC approval 12/YH/0330). This experimental work involves studies on genetically modified vectors already approved by the Health and Safety Executive (Azzouz_GMO_2006-07).

# Supplementary Information

# Acknowledgements

We thank Dr. Matthew Stopford for providing iAstrocytes and Hb9-GFP motor neurons; Dr. Adrian Higginbottom for providing 1321N1 human astrocytoma cells; Dr. Jason King for expertise on endocytosis; Professor Elisabeth Smythe for kindly providing Transferrin647 reagent; Dr. Colin Gray for support in confocal imaging and Dr. Jouni Takalo for guidance on statistical analysis. Imaging work was performed at the Wolfson Light Microscopy Facility using the Inverted Nikon Eclipse Ti microscope at the University of Sheffield with the grant code MR/K015753/1. We thank the Wolfson Foundation for their support in funding the Leica Confocal microscope at SiTraN. M Azzouz is supported by the European Research Council grant (ERC Advanced Award no. 294745), MRC DPFS award (129016), JPND-MRC (MR/V000470/1), ARUK award (ARUK-PG2018B-005), and CureAP4. L Marrone and M Azzouz are supported by Maddi Foundation and Spastic Paraplegia Foundation. L Marrone is further funded by a grant of the British Neuropathological Society. EF Smith was supported by a Motor Neurone Disease Association Prize Studentship (DeVos/Oct13/870-892 to KJ De Vos and AJ Grierson) and Motor Neurone Disease Association project grant (DEVOS/APR18/862-79 to KJ DeVos and AJ Grierson). KJ De Vos was supported by the Medical Research Council (MRC) (MR/S025979/1 and MR/M013251/1 to KJ De Vos). L Brasseur, L Bousset, and R Marroccella were supported by the European Commission Joint Programme on Neurodegenerative Diseases (JPND-TransPathND, ANR-17-JPCD-0002-02). The present work benefited from the electron microscopy facility of Imagerie-Gif (supported by ANR-10-INBS-04-01), and the Labex ANR-11-IDEX-0003-02. GM Hautbergue acknowledges support from the MRC New Investigator grant MR/R024162/1 and the Biotechnology and Biological Sciences Research Council grant BB/S005277/1. L Ferraiuolo is supported by the Academy of Medical Sciences (SBF002\1142) and the MRC (grant 1812144).

## Author Contributions

PM Marchi: conceptualization, data curation, formal analysis, validation, investigation, methodology, and writing—original draft.

L Marrone: validation, investigation, methodology, and writing—original draft, review, and editing.

L Brasseur: resources and methodology.

A Coens: methodology.

CP Webster: resources, methodology, and writing—review and editing.

L Bousset: resources, methodology, and writing—review and editing.

M Destro: resources.

EF Smith: resources and methodology.

CG Walther: software, validation, and methodology.

V Alfred: resources and methodology.

R Marroccella: resources and methodology.

EJ Graves: resources, data curation, formal analysis, and methodology.

D Robinson: resources, data curation, formal analysis, and methodology.

AC Shaw: resources and methodology.

LM Wan: resources and methodology.

AJ Grierson: resources.

SJ Ebbens: resources, software, and methodology.

KJ De Vos: resources and methodology.

GM Hautbergue: conceptualization, supervision, and writing—review and editing.

L Ferraiuolo: resources, supervision, investigation, methodology, and writing—review and editing.

R Melki: resources, methodology, and writing—review and editing.

M Azzouz: conceptualization, supervision, funding acquisition, project administration, and writing—review and editing.

## Conflict of Interest Statement

The authors declare that they have no conflict of interest.

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
