## [Reviewer comments · Life Science Alliance]

Life Science Alliance

C9orf72-derived poly-GA DPRs undergo endocytic uptake in iAstrocytes and spread to motor neurons

Paolo Marchi, Lara Marrone, Laurent Brasseur, Audrey Coens, Christopher Webster, Luc Bousset, Marco Destro, Emma Smith, Christa Walther, Victor Alfred, Raffaele Marroccella, Emily Graves, Darren Robinson, Allan Shaw, Lai Mei Wan, Andrew Grierson, Stephen Ebbens, Kurt De Vos, Guillaume Hautbergue, Laura Ferraiuolo, Ronald Melki, and Mimoun Azzouz
DOI: <https://doi.org/10.26508/lsa.202101276>

Corresponding author(s): Mimoun Azzouz, University of Sheffield and Ronald Melki, CEA, Institut François Jacob (MIRcen) and CNRS

Review Timeline:

Submission Date:	2021-10-26
Editorial Decision:	2021-11-23
Revision Received:	2022-04-07
Editorial Decision:	2022-04-29
Revision Received:	2022-05-03
Accepted:	2022-05-04

Transaction Report:

November 23, 2021

Re: Life Science Alliance manuscript #LSA-2021-01276-T

Prof. Mimoun Azzouz
University of Sheffield
Neuroscience
385a Glossop Rd, Broomhall
Sheffield S10 2HQ
United Kingdom

Dear Dr. Azzouz,

Thank you for submitting your manuscript entitled "C9orf72-derived poly-GA DPRs undergo endocytic uptake in iAstrocytes and spread to motor neurons" to Life Science Alliance. The manuscript was assessed by expert reviewers, whose comments are appended to this letter. We invite you to submit a revised manuscript addressing the Reviewer comments.

Thank you for this interesting contribution to Life Science Alliance. We are looking forward to receiving your revised manuscript.

Sincerely,

B. MANUSCRIPT ORGANIZATION AND FORMATTING:

Reviewer #1 (Comments to the Authors (Required)):

This manuscript by the Azzouz and Mimoun laboratories explores the biochemical properties and the spreading capacities of poly-GA, a major di-peptide repeat protein produced during C9ORF72 ALS and FTD. The authors first produce poly GA and poly PA (as control) and show that polyGA is able to oligomerize and fibrillize in vitro. PolyGA oligomers are efficiently taken up in various cell models, and the authors show that this occurs through dynamin dependent and independent endocytosis. Importantly, the authors then show that polyGA converge to the lysosome, leading to its dysfunction. Last, using iAstrocytes and motor neurons, the authors show that polyGA is capable of internalization in astrocytes, and can spread to motor neurons from these glial cells.

In all, this is a manuscript of outstanding quality, with each concept being replicated using different experimental approaches. This study very convincingly shows that polyGA is able to aggregate and spread from astrocytes to neurons in ALS, which might be of critical importance to understand C9ORF72 pathology.

My comments are very minor and can be addressed by modifications in the text of the manuscript without the obligate need of complementary experiments.

Minor comments:

- 1) Figure 4C claims to show that there is lysosomal dysfunction in primary neurons upon DPR exposure using RT-qPCR. This is a very indirect method to evaluate lysosomal dysfunction, and the conclusion of the authors on this specific point is questionable. I would advise the authors to either provide more direct evidence of lysosomal dysfunction, or, more simply, to remove these data from a revised version and tone down their claim of abnormal lysosomal function, which is already suggested by the rest of Figure 4.
- 2) The presentation of data is not always consistent from one panel to another. This is particularly the case in Figure 5C in which the experimental conditions are presented in various orders across subpanels i, ii and iii. It would be valuable to the reader if the authors present consistently their groups from one panel (and one figure) to the other.
- 3) The discussion section would benefit from a last paragraph stating clearly the limitations of the study in terms of disease relevance. It should be clearly acknowledged that these are in vitro experiments, and that DPR concentrations in vivo in patients could vary widely also due to the endogenous pattern of C9ORF72 expression. In addition, it should be mentioned that all experiments have been done at one short DPR length and that the influence of DPR and repeat expansion length on these cellular events remains to be determined.
- 4) On a very minor note, C9ORF72 being a human gene, it should be capitalized and italicized.

Reviewer #2 (Comments to the Authors (Required)):

In this manuscript, the authors have conducted a very interesting study showing ALS-linked C9orf72 RAN-translated DPR could be taken up through dynamin-dependent and independent endocytosis in astrocytes and spread to motor neurons. In detail, the authors show that Oligomeric and fibrillar poly-GA use distinct entry-routes in astrocytes, and poly-GA oligomers are taken up much faster than fibrillar poly-GA and Oligomeric poly-PA. Eventually, poly-GA induces lysosomal impairment and axonal swellings in neurons, suggesting a possible mechanism underlying the cellular pathogenicity of C9orf72 DPRs.

Overall, this manuscript contains logical experimental design and comprehensive cell biology and biochemical studies. The authors put effort to show the role of C9orf72 DPR-transmission in neurons and glial cells. It will not only help to understand the basic mechanism, but also shed light to the potential therapeutic applications bases on the findings in this paper. The manuscript could be of sufficient interest to the readers of Life Science Alliance, and there are several points need to be addressed by the authors before the publication:

1. The authors claimed that "Among the five different DPRs generated by RAN translation, poly-GA appears to be the most abundantly detected as well as one of the most toxic DPR species". However, poly-GR and poly-PR are usually considered as "the most" toxic DPR in vivo and in vitro - poly-GA can be considered as "the moderate" toxic DPR. Therefore, the author may

need to modify the introduction.

2. It is reasonable that the authors did not test poly-GR and poly-PR, two positive charge DPRs in this manuscript, since they display different properties and behaviors. However, it is useful to compare poly-GP, along with poly-GA and poly-PA.

3. Transcriptional levels of lysosomal genes such as ATP6V0E1, as well as the lysosomal size, do not reflect the overall "functional activity" of the lysosome. Therefore, the authors need to test at least one of the followings: (1) the lysosomal PH (acidification), or (2) the fusion with endosome/autophagosome, or (3) the degradation of classic substrates.

4. Overexposure of the lysotracker signal in Fig 4B.

5. Previous study showed that the spread of poly-GA could induce proteasomal impairment (Khosravi et al, 2020). Could the authors discuss or clarify the contribution of proteasomal VS lysosomal impairment in their model?

Reviewer #3 (Comments to the Authors (Required)):

In this manuscript, Marchi et al. examined the aggregation properties of poly-GA dipeptide repeat protein translated from G4C2 repeat expansions in the C9orf72 gene as well as its cellular uptake and cell-to-cell transmission. A number of previous studies have addressed the pathophysiological properties of poly-GA, including toxicity and cell-to-cell transmission. Some novel results are presented in this manuscript, but several important points have to be addressed to determine their pathophysiological significance.

Major points

- Cells were treated with poly-GA at a 0.5-1 μ M concentration. The authors mention in the Discussion that this range 'presumably recapitulates physiologically relevant conditions (Meeter et al, 2018)'. This assumption is based on the concentration of poly-GP measured in the CSF from expansion carriers. It is difficult to extrapolate from these data what cells will be exposed to in situ and 0.5-1 μ M may not reflect a pathophysiological situation. A wider range of concentrations should have been used.
- Following up from the above point, judging from the images presented, cells seem to have been overloaded with poly-GA; this is particularly apparent in Fig. 3A and in the axon shown in Fig. 4Ai/ii. Consequently, what is observed may not represent the physiological uptake mechanism that may be saturated under the experimental conditions used.
- Poly-GA fibrils are obtained after long incubation in vitro. Are some poly-GA oligomers still present at this stage? Some form of assessment/quantification is required. This is essential to interpret uptake results.
- Transmission of poly-GA from astrocytes to motor neurons is the most important result described in this manuscript and the authors should elaborate on this.
- Has the form of poly-GA released from astrocytes been investigated, as oligomers/fibrils may have been processed after uptake?
- Astrocyte toxicity has recently been shown to be mediated by saturated lipids (Guttenplan et al., 2021). Although no toxicity to motor neurons in co-cultures was observed, this is something that could be explored. For instance, does uptake of poly-GA oligomers/fibrils enhance release of saturated lipids?

Minor points

- The quantification provided in Fig. 2 refers to the percentage of cells with poly-GA aggregates, showing low uptake of poly-GA fibrils. However, individual aggregate-containing, fibril-treated, cells appear have a high load of poly-GA (e.g. Fig. 3A). An additional form of quantification should be provided.
- Fig. S2A is an important one and should be included in the main section of the manuscript.
- The authors may want to review their suggestion on the therapeutic benefit of blocking DPR uptake and secretion as they did not observe toxicity towards motor neurons in their experimental paradigm.
- Large parts of the Discussion are a detailed summary of the experiments performed. This could be reduced.
- On a very minor editing note, 'we' and 'our' are overused throughout the manuscript, using an indirect style more often would improve the text.

Referee Cross-Comments

I agree with the Reviewer #2's comment on functional activity of the lysosome. Since impairment of lysosome function by poly-GA is one of the main conclusions of the work, this should be investigated more thoroughly and the authors should follow Reviewer #2's suggestions for additional experiments.

RE: Life Science Alliance manuscript #LSA-2021-01276-T

# Response to Reviewer 1

Minor comments:

• **Figure 4C claims to show that there is lysosomal dysfunction in primary neurons upon DPR exposure using RT-qPCR. This is a very indirect method to evaluate lysosomal dysfunction, and the conclusion of the authors on this specific point is questionable. I would advise the authors to either provide more direct evidence of lysosomal dysfunction or, more simply, to remove these data from a revised version and tone down their claim of abnormal lysosomal function, which is already suggested by the rest of Figure 4.**

We agree with the Reviewer's comment in relation to the indirect evidence of lysosomal dysfunction provided in the previous Figure 4C. We therefore decided to move Figure 4C in the Supplementary panel (now Figure S7A). We have also performed additional lysosomal assays to look more directly at potential lysosomal dysfunction. Specifically, the protein levels of cathepsins L, B, and D were measured in primary neurons treated with poly-GA DPRs for 24 hours using immunofluorescence assays (now Figure S7B); we also quantified *in situ* lysosomal enzyme activities in primary neurons after 24 hours of poly-GA exposure (now Figure S7C and D); and used the galectin-3 immunofluorescence puncta assay to look for lysosomal damage in DPR-exposed HeLa cells (now Figure S7E and G), including LysoSensor Green to measure lysosomal pH (Figure S7F and H). To comment and discuss these new findings we included new paragraphs in the Results (page 9, rows 202-219) and Discussion (page 12, rows 291-299).

• **The presentation of data is not always consistent from one panel to another. This is particularly the case in Figure 5C, in which the experimental conditions are presented in various orders across subpanels i, ii and iii. It would be valuable to the reader if the authors presented their groups consistently from one panel (and one figure) to the other.**

We thank Reviewer 1 for this comment, we have now changed Figure 5C accordingly. We also added consistency for the subpanels' nomenclature using only letters (B, C, D, etc.) across all figures of the manuscript, we therefore eliminated subpanels named with (i), (ii), (iii) and the order of some panels was slightly rearranged to confer more clarity.

- **The discussion section would benefit from a last paragraph stating clearly the limitations of the study in terms of disease relevance. It should be clearly acknowledged that these are *in vitro* experiments and that DPR concentrations *in vivo* in patients could vary widely also due to the endogenous pattern of *C9ORF72* expression. In addition, it should be mentioned that all experiments have been done at one short DPR length and that the influence of DPR and repeat expansion length on these cellular events remains to be determined.**

We thank Reviewer 1 for this comment. We have now stated some of the limitations related to disease relevance in the discussion section. Specifically, we added the following section (in yellow) in the final part of the discussion; page 14, rows 342-348:

“It should be noted that findings in cell culture settings may not reflect the situation in animal models or humans: for example, DPR concentrations *in vivo* in patients may vary widely due to the endogenous pattern of *C9ORF72* expression, and the short DPR length used in our study cannot reflect the potential influence of longer DPR repeats on our described cellular events. However, our study remains informative for providing some fundamental insights into the biology related to poly-GA aggregation, binding, uptake and cell-to-cell propagation in glia and neurons.”

- **On a very minor note, *C9ORF72* being a human gene, it should be capitalized and italicized.**

Thank you for highlighting this point. This has been altered accordingly.

Response to Reviewer 2

• **The authors claimed that "Among the five different DPRs generated by RAN translation, poly-GA appears to be the most abundantly detected as well as one of the most toxic DPR species". However, poly-GR and poly-PR are usually considered as "the most" toxic DPR in vivo and in vitro - poly-GA can be considered as "the moderate" toxic DPR. Therefore, the author may need to modify the introduction.**

We appreciate Reviewer 2's comment as a quality point to improve our manuscript. We have added the following new section (in yellow) to the previous text on page 3 rows 50-56:

“Among the five different DPRs generated by RAN translation, the most toxic species are considered to be arginine-containing ones, namely poly-GR and poly-PR. These DPRs have been shown to alter the formation of membrane-less organelles such as stress granules or nucleoli (Lee *et al*, 2016; Lin *et al*, 2016; Tao *et al*, 2015; Zhang *et al*, 2018), cause mitochondrial dysfunction and DNA damage (Choi *et al*, 2019; Lopez-Gonzalez *et al*, 2016), and their expression is toxic in mice and in iPSC-derived cortical and motor neurons (Cook *et al*, 2020; Lopez-Gonzalez *et al*, 2016).”

• **It is reasonable that the authors did not test poly-GR and poly-PR, two positive charge DPRs in this manuscript, since they display different properties and behaviours. However, it is useful to compare poly-GP, along with poly-GA and poly-PA.**

Thank you for this comment. We have now included some data related to poly-GP DPRs in iAstrocytes. In particular, we exposed these cells to poly-GP for 24 hours and subsequently harvested the conditioned medium to measure APOJ protein levels by enzyme-linked immunosorbent assay (Figure S8B). The poly-GP assemblies were produced in Dr. Melki's lab and were extensively characterised in a previous study for molecular weight, secondary structure, and fluorescent labelling (Brasseur *et al*, 2020).

• **Transcriptional levels of lysosomal genes such as *ATP6V0E1*, as well as the lysosomal size, do not reflect the overall "functional activity" of the lysosome. Therefore, the authors need to test at least one of the following: (1) the lysosomal PH (acidification), or (2) the fusion with endosome/autophagosome, or (3) the degradation of classic substrates.**

We thank Reviewer 2 for this comment. We have performed additional lysosomal assays to look more directly at potential lysosomal dysfunction (now Figure S7). To comment and discuss these new findings we included new paragraphs in the Results (page 9, rows 202-219) and Discussion (page 12, rows 291-299). Please refer to our response to the first comment of Reviewer 1, where we briefly describe these additional lysosomal assays.

- **Overexposure of the lysotracker signal in Fig 4B.**

Thank you for emphasising this point. We highly value Reviewer 2's comment because overexposing cells to high laser power can be toxic and cause organelle movement changes (Stockley *et al*, 2017; Magidson & Khodjakov, 2013), but in our case only a small amount of saturated signal was collected and it only accumulated on larger lysosomes. Unfortunately, we had to increase the laser power of the microscope to detect the smallest lysosomes. However, lysosomal movement is highly present and dynamic, as shown in Movie S7, and our measured speed of Non-Colocalized Lysosomes (NCLs) corresponds to commonly reported values in the literature, such as 0.2µm/s (Bandyopadhyay *et al*, 2014; Szymanski *et al*, 2011). Because previous research has shown that increased lysosome diameter is associated with decreased diffusive lysosome motion (Bandyopadhyay *et al*, 2014), we believe that the slower speed of our Colocalized Lysosomes (CLs) can be attributed to their larger size/diameter as part of a true biological effect. Therefore, we believe that the signal overexposure on larger lysosomes had no significant effect on our functional read-outs.

- **Previous study showed that the spread of poly-GA could induce proteasomal impairment (Khosravi *et al*, 2020). Could the authors discuss or clarify the contribution of proteasomal vs lysosomal impairment in their model?**

We thank Reviewer 2 for this comment. We have now added a new section (in yellow) in the Discussion to address proteasome VS lysosomal impairment in our model (page 14, rows 332-341):

“Interestingly, in cell culture studies, the ability of poly-GA aggregates to transmit from cell to cell has been proven to disseminate proteasome inhibition and TDP-43 disease in neighbouring cells (Khosravi *et al*, 2020). Although the capacity of poly-GA to propagate *in vivo* in mammals has yet to be determined, poly-GA derived-proteasome impairment could be upstream of TDP-43 disease and be intertwined with impaired autophagy function caused by *C9ORF72* haploinsufficiency. Such proteasome and autophagy dysfunctions may exacerbate vulnerability not only in neurons but also in astrocytes, making these glial cells less effective in clearing pathological aggregates and in supporting neuronal homeostasis; as shown in the context of ALS *FIG4* mutations (Ferguson *et al*, 2009) and in a lysosomal storage disease study (Di Malta *et al*, 2012).”

Response to Reviewer 3

• Cells were treated with poly-GA at a 0.5-1 μ M concentration. The authors mention in the Discussion that this range 'presumably recapitulates physiologically relevant conditions (Meeter et al., 2018)'. This assumption is based on the concentration of poly-GP measured in the CSF from expansion carriers. It is difficult to extrapolate from these data what cells will be exposed to in situ, and 0.5-1 μ M may not reflect a pathophysiological situation. A wider range of concentrations should have been used.

We agree that the 0.5-1 μ M range of poly-GA has a hard time holding any relation to the human study that we mentioned, which indeed measured poly-GP levels in the CSF. We therefore removed the part “*presumably recapitulates physiologically relevant conditions (Meeter et al., 2018)*” from our manuscript. We appreciate Reviewer 3 bringing this to our attention.

• Following up from the above point, judging from the images presented, cells seem to have been overloaded with poly-GA; this is particularly apparent in Fig. 3A and in the axon shown in Fig. 4Ai/ii. Consequently, what is observed may not represent the physiological uptake mechanism that may be saturated under the experimental conditions used.

We thank Reviewer 3 for bringing up this very essential point. This issue is related to our initial claims that the concentrations we used are physiological. We toned down this claim as indicated above.

Nonetheless, in the experiment of the previous Fig. 3A (now Fig. 4A), the presence of DPR aggregates in cells varies greatly, with some cells having a high aggregate load and others having undetectable or low levels. The highly heterogeneous nature of our dataset suggests that the cells are not saturated and might be supported by the idea that aggregation can begin in one “rogue cell” or extracellular area and then spread within neighbouring cells/regions; this idea is under discussion for several other amyloid diseases (Polymenidou & Cleveland, 2012; Eisele *et al*, 2015; Prusiner, 2012). The high heterogeneity of DPR levels led us to use an unbiased algorithm to measure the average fluorescent values of Atto550-DPR signal per cell, and we had to log-transform (\log_{10}) the values for better graph visualization. The images in the previous Fig. 3A are only representative of the final fold-change difference shown in the graph.

In the experiment of the previous Fig. 4Ai/ii (now Fig. 5A), we're imaging large axonal swellings which are highly frequent structures at the microfluidic grooves. These swellings, also called spheroids, usually contain elevated quantities of aggregation-prone proteins as reported in a number of studies in relation to neurodegenerative diseases (Wirhns *et al*, 2006; Hadano *et al*, 2010). Thus, we are confident that our result represents a real situation in which axonal swellings are highly enriched with poly-GA DPRs and therefore the signal appears to be saturated/overloaded due to i) high

concentration of DPRs in these structures and ii) increased sensitivity and magnification of airyscan confocal mode with 63x lens.

As we state in the Discussion, 0.5-1 μM range of poly-GA was used in all experiments; no higher concentrations were used. I hope we were able to clarify these essential aspects of the “nature” of our datasets, and we thank Reviewer 3 for bringing up this important point to our attention.

Regarding the considerations of Reviewer 3 on the uptake mechanism representing a physiological situation, please refer to our response to the third comment of Reviewer 1, where we briefly describe the limitations of our cell culture findings in reflecting phenotypes in animal models and humans.

• Poly-GA fibrils are obtained after long incubation *in vitro*. Are some poly-GA oligomers still present at this stage? Some form of assessment/quantification is required. This is essential to interpret uptake results.

We thank the reviewer for bringing this up. We previously assessed the fibrillization kinetics by SDS-PAGE by centrifugation at 100 000g and measurement of disappearance over time of monomeric DPRs (Brasseur *et al*, 2020). We indicated in the methods section (page 18, line 432-433), poly-GA fibrils are spun at 100 000g for 30 min after labelling. This step is actually performed to remove the unreacted dye but also to get rid of unassembled polypeptides. We therefore changed the sentence to clarify this issue. “Unreacted NHS-dye and non-fibrillar polypeptide was removed by centrifugation (100’000g, 30 minutes, 4°C)”.

• Transmission of poly-GA from astrocytes to motor neurons is the most important result described in this manuscript, and the authors should elaborate on this.

We thank Reviewer 3 for this comment. We further elaborated the implications of astrocyte-to-motor neuron propagation in the Discussion section. Specifically, we have now added the new following part (in yellow); at page 13-14, rows 326-341:

“Our data propose an important role for astrocytes in the transmission of *C9ORF72*-derived DPRs to neighbouring motor neurons. Whether this mechanism could constitute a “non-cell autonomous” contributor in *C9ORF72*-ALS/FTD neuropathology remains to be fully elucidated. Although DPR *post-mortem* inclusions have been detected to a lesser extent in glial cells than in neurons, thus receiving less attention, we speculate that DPR aggregation in glia may still go undetected earlier in disease and play a role in disease pathogenesis. For instance, astrocytes could act as hubs for the continuous internalisation and redistribution of DPRs. Interestingly, in cell culture studies, the ability of poly-GA aggregates to transmit from cell to cell has been proven to disseminate proteasome inhibition and TDP-43 disease in neighbouring cells (Khosravi *et al*, 2020). Although the capacity of poly-GA to propagate *in vivo* in mammals has yet to be determined, poly-GA derived-proteasome

impairment could be upstream of TDP-43 disease and be intertwined with impaired autophagy function caused by *C9ORF72* haploinsufficiency.”

• Has the form of poly-GA released from astrocytes been investigated, as oligomers/fibrils may have been processed after uptake?

We thank the reviewer for bringing this issue up. Our experiments do not allow us to directly answer this concern. We previously showed the fibrillar and oligomeric forms of the DPRs we used are highly stable over time. This is true in buffer and in culture media. We also showed they resist limited proteolysis. The species we used are fluorescently labelled on the glycine amine (N-terminal amino acid residue of the peptide generated after TEV cleavage of the polypeptides schematized in Fig S1) and the unique lysine residue within the V5 tag, or two lysine residues within the FLAG tag, with NHS-dyes. Their processing within the cell should result in the release and dilution of the fluorophores. This is not what we observe. Thus, it is reasonable to consider they are not processed significantly given that their fluorescence appears not to change significantly.

• Astrocyte toxicity has recently been shown to be mediated by saturated lipids (Guttenplan *et al*, 2021). Although no toxicity to motor neurons in co-cultures was observed, this is something that could be explored. For instance, does uptake of poly-GA oligomers/fibrils enhance the release of saturated lipids?

We very much appreciate this suggestion by Reviewer 3, which brings into the manuscript a very interesting experiment. Indeed, an increase in apolipoprotein J (APOJ) (also known as clusterin) in the conditioned medium of astrocytes has been linked to astrocyte-mediated toxicity. In particular, APOJ is secreted into the conditioned medium in lipidated lipoparticles, which have been shown in Gutteplan and colleagues' study to constitute a toxicity feature of reactive astrocytes. As a result, we used an enzyme-linked immunosorbent assay (ELISA) to determine the protein levels of APOJ in the conditioned medium of either untreated or treated iAstrocytes for 24 hours with various recombinant DPRs (poly-GA fibrils, poly-GA oligomers, poly-PA oligomers, poly-GP oligomers). The newly added Figure S8B depicts the quantification of APOJ levels. We added new paragraphs to the Results (page 10, rows 239-244) and Discussion (page 13, rows 320-322) to comment on and discuss these new findings.

Minor comments:

• The quantification provided in Fig. 2 refers to the percentage of cells with poly-GA aggregates, showing low uptake of poly-GA fibrils. However, individual aggregate-containing, fibril-treated,

cells appear to have a high load of poly-GA (e.g. Fig. 3A). An additional form of quantification should be provided.

We thank Reviewer 3 for bringing up another essential point regarding our datasets. The different microscopes used for these figures (previous Fig. 2 vs Fig. 3A) strongly influence the detection of DPRs.

In Figure 2, we used a PerkinElmer high-throughput system with a 1.1NA 40x lens to image what we have referred to as "largely visible DPR aggregates" (Result section, row 110 - in previous version). Considering the use of ATTO550 dye and a 1.1NA lens, the lateral resolution achievable with this system is >320 nm and therefore only largely visible aggregates (primarily large aggresomes) are successfully detected and considered for the analysis of percent cellular uptake in a high-throughput manner.

Conversely, in Fig. 3A, our purpose was to verify the cellular internalisation of DPRs in vimentin-stained cells using Z-stack airyscan confocal microscopy, orthogonal views and 3D-volume rendering. This experiment was set out to provide a yes-or-no answer on the occurrence of DPR internalisation in the cells rather than a high-throughput quantification. Considering the use of ATTO550 dye and a 1.4NA lens with airyscan mode (Zeiss LSM800), the lateral resolution achievable with this system is ~120-140 nm.

Because of these two different imaging modalities which were utilised for two different experimental purposes, we do not encourage to compare DPR uptake between Fig. 2 and Fig. 3A as it would produce

incomparable results during data interpretation. We ask permission to eventually include these comments in a Material and Method section to better clarify these important aspects that Reviewer 3 brought to our attention.

• Fig. S2A is an important one and should be included in the main section of the manuscript.

We thank Reviewer 3 for this comment. We have now moved the previous Figure S2A to the main figure panel and is now named Figure 3C. To give more emphasis to these findings we slightly expanded the previous version by adding a few rows in the Results (now page 6-7, rows 137-145) – in yellow:

“ Interestingly, our model suggests that this active force could be descriptive of microtubule-mediated transport, as reported in a previous study (Van Den Heuvel *et al*, 2007). To obtain the experimental confirmation, we exposed healthy iAstrocytes to poly-GA (24h) and subsequently subjected the cells to a 30 minute-treatment with the microtubule de-polymerising agent nocodazole (30 μ M). Cells were fixed and stained with anti-tubulin antibody to confirm microtubule de-polymerization. Our results show that the microtubule de-polymerising agent nocodazole induces cellular relocation of poly-GA

DPRs (Fig 3C), hence suggesting that a certain fraction of poly-GAs undergoes microtubule-mediated transport after cell entry.”

- **The authors may want to review their suggestion on the therapeutic benefit of blocking DPR uptake and secretion as they did not observe toxicity towards motor neurons in their experimental paradigm.**

Thank you for highlighting this point. Indeed, the pathogenic potential of DPR cell propagation observed in our cell culture findings remains to be determined. We have now deleted the following sentence from our manuscript:

“We envision that future therapies in the context of *C9ORF72*-ALS could benefit from blocking DPR uptake- and secretion-routes not only in neuronal cells but also in glia.”

- **Large parts of the Discussion are a detailed summary of the experiments performed. This could be reduced.**

Thank you for this comment. We have now reduced the length of the discussion section by removing some descriptive results that, as Reviewer 3 pointed out, were also present in the Results section and thus constituted a repetition. We had to add a few discussion points about the new findings (for example, lysosomal assays and APOJ-ELISA) and Reviewers' comments (e.g., discussion of proteasomal vs lysosomal impairment). Despite the addition of these new parts, the length of the new Discussion has remained roughly the same as the previous one (only 5 rows more).

- **On a very minor editing note, 'we' and 'our' are overused throughout the manuscript; using an indirect style more often would improve the text.**

Thank you for your suggestion. This section has now been addressed, and several repetitions of "we" and "our" in our manuscript have been removed. We agree that an indirect style benefits the text.

Referee Cross-Comments:

- **I agree with Reviewer #2's comment on the functional activity of the lysosome. Since impairment of lysosome function by poly-GA is one of the main conclusions of the work, this should be investigated more thoroughly, and the authors should follow Reviewer #2's suggestions for additional experiments.**

Thank you for this comment. We have performed additional lysosomal assays to look more directly at potential lysosomal dysfunction (now Figure S7). To comment and discuss these new findings we included new paragraphs in the Results (page 9, rows 202-219) and Discussion (page 12, rows 291-299). Please refer to our response to the first comment of Reviewer 1, where we briefly describe these additional lysosomal assays.

References

- Bandyopadhyay D, Cyphersmith A, Zapata JA, Kim YJ & Payne CK (2014) Lysosome Transport as a Function of Lysosome Diameter. *PLoS One* 9: e86847
- Brasseur L, Coens A, Waeytens J, Melki R & Bousset L (2020) Dipeptide repeat derived from C9orf72 hexanucleotide expansions forms amyloids or natively unfolded structures in vitro. *Biochem Biophys Res Commun*: 0–6
- Choi SY, Lopez-Gonzalez R, Krishnan G, Phillips HL, Li AN, Seeley WW, Yao WD, Almeida S & Gao FB (2019) C9ORF72-ALS/FTD-associated poly(GR) binds Atp5a1 and compromises mitochondrial function in vivo. *Nat Neurosci* 22: 851–862
- Cook CN, Wu Y, Odeh HM, Gendron TF, Jansen-West K, del Rosso G, Yue M, Jiang P, Gomes E, Tong J, *et al* (2020) C9orf72 poly(GR) aggregation induces TDP-43 proteinopathy. *Sci Transl Med* 12
- Eisele YS, Monteiro C, Fearn C, Encalada SE, Wiseman RL, Powers ET & Kelly JW (2015) Targeting protein aggregation for the treatment of degenerative diseases. *Nat Rev Drug Discov* 14: 759–780
- Ferguson CJ, Lenk GM & Meisler MH (2009) Defective autophagy in neurons and astrocytes from mice deficient in PI(3,5)P2. *Hum Mol Genet* 18: 4868–4878
- Guttenplan KA, Weigel MK, Prakash P, Wijewardhane PR, Hasel P, Rufen-Blanchette U, Münch AE, Blum JA, Fine J, Neal MC, *et al* (2021) Neurotoxic reactive astrocytes induce cell death via saturated lipids. *Nature* 599: 102–107
- Hadano S, Otomo A, Kunita R, Suzuki-Utsunomiya K, Akatsuka A, Koike M, Aoki M, Uchiyama Y, Itoyama Y & Ikeda J-E (2010) Loss of ALS2/Alsin Exacerbates Motor Dysfunction in a SOD1H46R-Expressing Mouse ALS Model by Disturbing Endolysosomal Trafficking. *PLoS One* 5: e9805
- Van Den Heuvel MGL, Bolhuis S & Dekker C (2007) Persistence length measurements from stochastic single-microtubule trajectories. *Nano Lett* 7: 3138–3144
- Khosravi B, LaClair KD, Riemenschneider H, Zhou Q, Frottin F, Mareljic N, Czuppa M, Farny D, Hartmann H, Michaelsen M, *et al* (2020) Cell-to-cell transmission of C9orf72 poly-(Gly-Ala) triggers key features of ALS / FTD. *EMBO J*: 1–19
- Lee KH, Zhang P, Kim HJ, Mitrea DM, Sarkar M, Freibaum BD, Cika J, Coughlin M, Messing J, Molliex A, *et al* (2016) C9orf72 Dipeptide Repeats Impair the Assembly, Dynamics, and

Function of Membrane-Less Organelles. *Cell* 167: 774-788.e17

- Lin Y, Mori E, Kato M, Xiang S, Wu L, Kwon I & McKnight SL (2016) Toxic PR Poly-Dipeptides Encoded by the C9orf72 Repeat Expansion Target LC Domain Polymers. *Cell* 167: 789-802.e12
- Lopez-Gonzalez R, Lu Y, Gendron TF, Karydas A, Tran H, Yang D, Petrucelli L, Miller BL, Almeida S & Gao FB (2016) Poly(GR) in C9ORF72-Related ALS/FTD Compromises Mitochondrial Function and Increases Oxidative Stress and DNA Damage in iPSC-Derived Motor Neurons. *Neuron* 92: 383–391
- Magidson V & Khodjakov A (2013) Circumventing photodamage in live-cell microscopy. In *Methods in Cell Biology* pp 545–560. Academic Press Inc.
- Di Malta C, Fryer JD, Settembre C & Ballabio A (2012) Astrocyte dysfunction triggers neurodegeneration in a lysosomal storage disorder. *Proc Natl Acad Sci* 109
- Polymenidou M & Cleveland DW (2012) Prion-like spread of protein aggregates in neurodegeneration. *J Exp Med* 209: 889–893
- Prusiner SB (2012) A Unifying Role for Prions in Neurodegenerative Diseases. *Science* (80-) 336: 1511–1513
- Stockley JH, Evans K, Matthey M, Volbracht K, Agathou S, Mukanowa J, Burrone J & Káradóttir RT (2017) Surpassing light-induced cell damage in vitro with novel cell culture media. *Sci Rep* 7: 1–11
- Szymanski CJ, Humphries, IV WH & Payne CK (2011) Single particle tracking as a method to resolve differences in highly colocalized proteins. *Analyst* 136: 3527
- Tao Z, Wang H, Xia Q, Li K, Li K, Jiang X, Xu G, Wang G & Ying Z (2015) Nucleolar stress and impaired stress granule formation contribute to C9orf72 RAN translation-induced cytotoxicity. *Hum Mol Genet* 24: 2426–41
- Wirhlich O, Weis J, Szczygielski J, Multhaup G & Bayer TA (2006) Axonopathy in an APP/PS1 transgenic mouse model of Alzheimer’s disease. *Acta Neuropathol* 111: 312–319
- Zhang YJ, Gendron TF, Ebbert MTW, O’Raw AD, Yue M, Jansen-West K, Zhang X, Prudencio M, Chew J, Cook CN, *et al* (2018) Poly(GR) impairs protein translation and stress granule dynamics in C9orf72-associated frontotemporal dementia and amyotrophic lateral sclerosis. *Nat Med* 24: 1136–1142 doi:10.1038/s41591-018-0071-1

April 29, 2022

RE: Life Science Alliance Manuscript #LSA-2021-01276-TR

Prof. Mimoun Azzouz
University of Sheffield
Neuroscience
385a Glossop Rd, Broomhall
Sheffield S10 2HQ
United Kingdom

Dear Dr. Azzouz,

Thank you for submitting your revised manuscript entitled "C9orf72-derived poly-GA DPRs undergo endocytic uptake in iAstrocytes and spread to motor neurons". We would be happy to publish your paper in Life Science Alliance pending final revisions necessary to meet our formatting guidelines.

- please clearly specify the DPR concentration used in all relevant parts of the Results section and in the figure legends
- please add ORCID ID for secondary corresponding author; they should have received instructions on how to do so
- please make sure the author order in the system and in the manuscript match
- please introduce your Figure Panels in alphabetical order in the legend for Figure S7
- please add an ethics statement to include approval for animal use, and who granted the approval

A. FINAL FILES:

B. MANUSCRIPT ORGANIZATION AND FORMATTING:

**Submission of a paper that does not conform to Life Science Alliance guidelines will delay the acceptance of your

manuscript.**

The license to publish form must be signed before your manuscript can be sent to production. A link to the electronic license to publish form will be sent to the corresponding author only. Please take a moment to check your funder requirements.

Sincerely,

Reviewer #1 (Comments to the Authors (Required)):

The authors addressed adequately my comments.
On a minor note, Ms Audrey Coens is now listed as author, and should not be acknowledged, and the authors contribution section should thus be adequately modified.
I congratulate the authors for their important work.

Reviewer #2 (Comments to the Authors (Required)):

The authors have fully addressed all my previous concerns, and the manuscript is now suitable for publication.

Reviewer #3 (Comments to the Authors (Required)):

My main concern was the significance of the results presented, especially due to the concentration of poly-GA used. The authors have addressed the issue by simply acknowledging it and deleting a comment that 0.5-1 µM poly-GA was presumably physiological. This does not solve the problem; a wider range of concentration must be used. In fact, I agree with the authors' conclusion that:

"It should be noted that findings in cell culture settings may not reflect the situation in animal models or humans: for example, DPR concentrations in vivo in patients may vary widely due to the endogenous pattern of C9ORF72 expression, and the short DPR length used in our study cannot reflect the potential influence of longer DPR repeats on our described cellular events."

Consequently, the work does not have much of a pathophysiologically meaningful conclusion.

May 4, 2022

RE: Life Science Alliance Manuscript #LSA-2021-01276-TRR

Prof. Mimoun Azzouz
University of Sheffield
Neuroscience
385a Glossop Rd, Broomhall
Sheffield S10 2HQ
United Kingdom

Dear Dr. Azzouz,

Thank you for submitting your Research Article entitled "C9orf72-derived poly-GA DPRs undergo endocytic uptake in iAstrocytes and spread to motor neurons". It is a pleasure to let you know that your manuscript is now accepted for publication in Life Science Alliance. Congratulations on this interesting work.

DISTRIBUTION OF MATERIALS:

Again, congratulations on a very nice paper. I hope you found the review process to be constructive and are pleased with how the manuscript was handled editorially. We look forward to future exciting submissions from your lab.

Sincerely,
